# Human complex exploration strategies are enriched by noradrenaline-modulated heuristics

**Magda Dubois[1,2]\*, Johanna Habicht[1,2], Jochen Michely[1,2,3], Rani Moran[1,2], Ray J Dolan[1,2], Tobias U Hauser[1,2]\***

[1]Max Planck UCL Centre for Computational Psychiatry and Ageing Research, London, United Kingdom; [2]Wellcome Trust Centre for Neuroimaging, University College London, London, United Kingdom; [3]Department of Psychiatry and Psychotherapy, Charité – Universitätsmedizin Berlin, Berlin, Germany

**Abstract** An exploration-exploitation trade-off, the arbitration between sampling a lesser-known against a known rich option, is thought to be solved using computationally demanding exploration algorithms. Given known limitations in human cognitive resources, we hypothesised the presence of additional cheaper strategies. We examined for such heuristics in choice behaviour where we show this involves a value-free random exploration, that ignores all prior knowledge, and a novelty exploration that targets novel options alone. In a double-blind, placebo-controlled drug study, assessing contributions of dopamine (400 mg amisulpride) and noradrenaline (40 mg propranolol), we show that value-free random exploration is attenuated under the influence of propranolol, but not under amisulpride. Our findings demonstrate that humans deploy distinct computationally cheap exploration strategies and that value-free random exploration is under noradrenergic control.

**\*For correspondence:**
magda.dubois.18@ucl.ac.uk (MD);
t.hauser@ucl.ac.uk (TUH)

## Introduction

Chocolate, Toblerone, spinach, or hibiscus ice-cream? Do you go for the flavour you like the most (chocolate), or another one? In such an exploration-exploitation dilemma, you need to decide whether to go for the option with the highest known subjective value (exploitation) or opt instead for less known or valued options (exploration) so as to not miss out on possibly even higher rewards. In the latter case, you can opt to either choose an option that you have previously enjoyed (Toblerone), an option you are curious about because you do not know what to expect (hibiscus), or even an option that you have disliked in the past (spinach). Depending on your exploration strategy, you may end up with a highly disappointing ice cream encounter, or a life-changing gustatory epiphany.

A common approach to the study of complex decision making, for example an exploration-exploitation trade-off, is to take computational algorithms developed in the field of artificial intelligence and test whether key signatures of these are evident in human behaviour. This approach has revealed humans use strategies that reflect an implementation of computationally demanding exploration algorithms (*Gershman, 2018*; *Schulz and Gershman, 2019*). One such strategy, directed exploration, involves awarding an 'information bonus' to choice options, a bonus that scales with uncertainty. This is captured in algorithms such as the Upper Confidence Bound (UCB) (*Auer, 2003*; *Carpentier et al., 2011*) and leads to an exploration of choice options the agent knowns little about (*Gershman, 2018*; *Schwartenbeck et al., 2019*) (e.g. the hibiscus ice-cream). An alternative strategy, sometimes termed 'random' exploration, is to induce stochasticity after value computations in the decision process. This can be realised using a fixed parameter as a source of stochasticity, such as a softmax temperature parameter (*Daw et al., 2006*; *Wilson et al., 2014*), which can be

combined with the UCB algorithm (*Gershman, 2018*). Alternatively, one can use a dynamic source of stochasticity, such as in Thompson sampling (*Thompson, 1933*), where stochasticity adapts to an uncertainty about choice options. This exploration is essentially a more sophisticated, uncertainty-driven, version of a softmax. By accounting for stochasticity when comparing choice options' expected values, in effect choosing based on both uncertainty and value, these exploration strategies increase the likelihood of choosing 'good' options that are only slightly less valuable than the best (e.g. the Toblerone ice-cream if you are a chocolate lover).

The above processes are computationally demanding, especially when facing real-life multiple-alternative decision problems (*Daw et al., 2006*; *Cohen et al., 2007*; *Cogliati Dezza et al., 2019*). Human cognitive resources are constrained by capacity limitations (*Papadopetraki et al., 2019*), metabolic consumption (*Zénon et al., 2019*), but also because of resource allocation to parallel tasks (e.g.*Wahn and König, 2017*; *Marois and Ivanoff, 2005*). This directly relates to an agents' motivation to perform a given task (*Papadopetraki et al., 2019*; *Botvinick and Braver, 2015*; *Fröböse et al., 2020*), as increasing an information demand in one process automatically reduces its availability for others (*Zénon et al., 2019*). In real-world highly dynamic environments, this arbitration is critical as humans need to maintain resources for alternative opportunities (i.e. flexibility; *Papadopetraki et al., 2019*; *Kool et al., 2010*; *Cools, 2015*). This accords with previous studies showing humans are demand-avoidant (*Kool et al., 2010*; *Fröböse and Cools, 2018*) and suggests that exploration computations tend to be minimised. Here, we examine the explanatory power of two additional computationally less costly forms of exploration, namely value-free random exploration and novelty exploration.

Computationally, the least resource demanding way to explore is to ignore all prior information and to choose entirely randomly, de facto assigning the same probability to all options. Such 'value-free' random exploration, as opposed to the two previously considered 'value-based' random explorations (for simulations comparing their effects *Figure 1—figure supplement 2*) that add stochasticity during choice value computation, forgoes any costly computation (i.e. value mean and uncertainty), known as an $\epsilon$-greedy algorithmic strategy in reinforcement learning (*Sutton and Barto, 1998*). Computational efficiency, however, comes at the cost of sub-optimality due to occasional selection of options of low expected value (e.g. the repulsive spinach ice cream).

Despite its sub-optimality, value-free random exploration has neurobiological plausibility. Of relevance in this context is a view that exploration strategies depend on dissociable neural mechanisms (*Zajkowski et al., 2017*). Influences from noradrenaline and dopamine are plausible candidates in this regard based on prior evidence (*Cohen et al., 2007*; *Hauser et al., 2016*). Amongst other roles (such as memory [*Sara et al., 1994*], or energisation of behaviour [*Varazzani et al., 2015*; *Silvetti et al., 2018*]), the neuromodulator noradrenaline has been ascribed a function of indexing uncertainty (*Silvetti et al., 2013*; *Yu and Dayan, 2005*; *Nassar et al., 2012*) or as acting as a 'reset button' that interrupts ongoing information processing (*David Johnson, 2003*; *Bouret and Sara, 2005*; *Dayan and Yu, 2006*). Prior experimental work in rats shows boosting noradrenaline leads to more value-free-random-like random behaviour (*Tervo et al., 2014*), while pharmacological manipulations in monkeys indicates reducing noradrenergic activity increases choice consistency (*Jahn et al., 2018*).

In human pharmacological studies, interpreting the specific function of noradrenaline on exploration strategies is problematic as many drugs, such as atomoxetine (e.g. *Warren et al., 2017*), impact multiple neurotransmitter systems. Here, to avoid this issue, we chose the highly specific $\beta$-adrenoceptor antagonist propranolol, which has only minimal impact on other neurotransmitter systems (*Fraundorfer et al., 1994*; *Hauser et al., 2019*). Using this neuromodulator, we examine whether signatures of value-free random exploration are impacted by administration of propranolol.

An alternative computationally efficient exploration heuristic to random exploration is to simply choose an option not encountered previously, which we term novelty exploration. Humans often show novelty seeking (*Bunzeck et al., 2012*; *Wittmann et al., 2008*; *Gershman and Niv, 2015*; *Stojić et al., 2020*), and this strategy can be used in exploration as implemented by a low-cost version of the UCB algorithm. Here, a novelty bonus (*Krebs et al., 2009*) is added if a choice option has not been seen previously (i.e. it does not have to rely on precise uncertainty estimates). The neuromodulator dopamine is implicated not only in exploration in general (*Frank et al., 2009*), but also in signalling such types of novelty bonuses, where evidence indicates a role in processing and exploring novel and salient states (*Wittmann et al., 2008*; *Bromberg-Martin et al., 2010*;

*Costa et al., 2014*; *Düzel et al., 2010*; *Iigaya et al., 2019*). Although pharmacological dopaminergic studies in humans have demonstrated effects on exploration as a whole (*Kayser et al., 2015*), they have not identified specific exploration strategies. Here, we used the highly specific D2/D3 antagonist amisulpride, to disentangle the specific role of dopamine and noradrenaline on different exploration strategies.

Thus, in the current study, we examine the contributions of value-free random exploration and novelty exploration in human choice behaviour. We developed a novel exploration task combined with computational modeling to probe the contributions of noradrenaline and dopamine. Under double-blind, placebo-controlled, conditions, we assessed the impact of two antagonists with high affinity and specificity for either dopamine (amisulpride) or noradrenaline (propranolol), respectively. Our results provide evidence that both exploration heuristics supplement computationally more demanding exploration strategies, and that value-free random exploration is particularly sensitive to noradrenergic modulation, with no effect of amisulpride.

## Results

### Probing the contributions of heuristic exploration strategies

We developed a novel multi-round three-armed bandit task (*Figure 1*; bandits depicted as trees), enabling us to assess the contributions of value-free random exploration and novelty exploration in addition to Thompson sampling and UCB (combined with a softmax). In particular, we exploited the fact that both heuristic strategies make specific predictions about choice patterns. The novelty exploration assigns a 'novelty bonus' only to bandits for which subjects have no prior information, but not to other bandits. This can be seen as a low-resolution version of UCB, which assigns a bonus to all choice options proportionally to how informative they are, in effect a graded bonus which scales to each bandit's uncertainty. Thus, to capture this heuristic, we manipulated the amount of prior information with bandits carrying only little information (i.e. 1 vs 3 initial samples) or no information (0 initial samples). A high novelty exploration predicts a higher frequency of selecting the novel option (*Figure 1f*). This is in contrast to high exploration using other strategies which does not predict such a strong effect on the novel option (*Figure 1—figure supplement 5*).

Value-free random exploration, captured here by $\epsilon$-greedy, predicts that all prior information is discarded entirely and that there is equal probability attached to all choice options. This strategy is distinct from other exploration strategies as it is likely to choose bandits known to be substantially worse than the other bandits. Thus, a high value-free random exploration predicts a higher frequency of selecting the low-value option (*Figure 1e*), whereas high exploration using other strategies does not predict such effect (*Figure 1—figure supplement 3*). A second prediction is that choice consistency, across repeated trials, is directly affected by value-free random exploration, in particular by comparison to other more deterministic exploration strategies (e.g. directed exploration) that are value-guided and thus will consistently select the most informative and valuable options. Given that value-free random exploration splits its choice probability equally (i.e. 33.3% of choosing any bandit out of the three displayed), an increase in such exploration predicts a lower likelihood of choosing the same bandit again, even under identical choice options (*Figure 1e*). This contrasts to other strategies that make consistent exploration predictions (e.g. UCB would consistently explore the choice option that carries a high information bonus; *Figure 1—figure supplement 4*).

We generated bandits from four different generative processes (*Figure 1c*) with distinct sample means (but a fixed sampling variance) and number of initial samples (i.e. samples shown at the beginning of a trial for this specific bandit). Subjects were exposed to these bandits before making their first draw. The 'certain-standard bandit' and the (less certain) 'standard bandit' were bandits with comparable means but varying levels of uncertainty, providing either three or one initial samples (depicted as apples; similar to the horizon task [*Wilson et al., 2014*]). The 'low-value bandit' was a bandit with one initial sample from a substantially lower generative mean, thus appealing to a value-free random exploration strategy alone. The last bandit, with a mean comparable with that of the standard bandits, was a 'novel bandit' for which no initial sample was shown, primarily appealing to a novelty exploration strategy (cf. Materials and methods for a full description of bandit generative processes). To assess choice consistency, all trials were repeated once. In the pilot experiments (data not shown), we noted some exploration strategies tended to overshadow other strategies. To

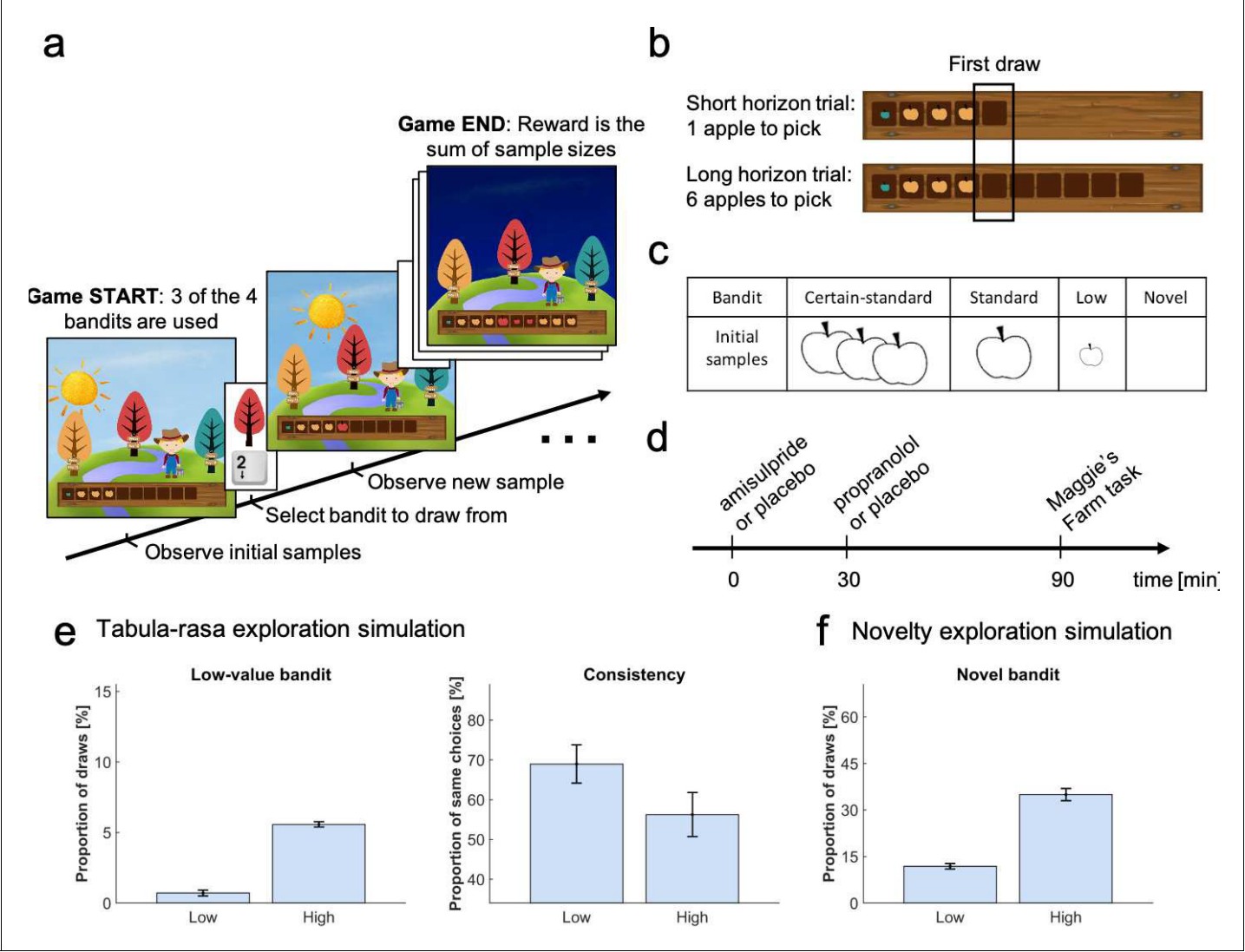

**Figure 1.** Study design. In the Maggie's farm task, subjects had to choose from three bandits (depicted as trees) to maximise an outcome (sum of reward). The rewards (apple size) of each bandit followed a normal distribution with a fixed sampling variance. (**a**) At the beginning of each trial, subjects were provided with some initial samples on the wooden crate at the bottom of the screen and had to select which bandit they wanted to sample from next. (**b**) Depending the condition, they could either perform one draw (short horizon) or six draws (long horizon). The empty spaces on the wooden crate (and the sun's position) indicated how many draws they had left. The first draw in both conditions was the main focus of the analysis. (**c**) In each trial, three bandits were displayed, selected from four possible bandits, with different generative processes that varied in terms of their sample mean and number of initial samples (i.e. samples shown at the beginning of a trial). The 'certain-standard bandit' and the 'standard bandit' had comparable means but different levels of uncertainty about their expected mean: they provided three and one initial sample, respectively; the 'low-value bandit' had a low mean and displayed one initial sample; the 'novel bandit' did not show any initial sample and its mean was comparable with that of the standard bandits. (**d**) Prior to the task, subjects were administered different drugs: 400 mg amisulpride that blocks dopaminergic D2/D3 receptors, 40 mg propranolol to block noradrenergic β-receptors, and inert substances for the placebo group. Different administration times were chosen to comply with the different drug pharmacokinetics (placebo matching the other groups' administration schedule). (**e**) Simulating value-free random behaviour with a low vs high model parameter ($\epsilon$) in this task shows that in a high regime, agents choose the low-value bandit more often (left panel; mean ± 1 SD) and are less consistent in their choices when facing identical choice options (right panel). (**f**) Novelty exploration exclusively promotes choosing choice options for which subjects have no prior information, captured by the 'novel bandit' in our task. For details about simulations cf. Materials and methods. For details about the task display *Figure 1—figure supplement 1*. For simulations of different exploration strategies and their impact of different bandits *Figure 1—figure supplement 2–5*.

The online version of this article includes the following figure supplement(s) for figure 1:

**Figure supplement 1.** Visualisation of the nine different sizes that the apples could take.

**Figure supplement 2.** Comparison of value-based (softmax) and value-free ($\epsilon$-greedy) random exploration.

*Figure 1 continued on next page*

*Figure 1 continued*

**Figure supplement 3.** Simulation illustrations of high and low exploration on the frequency of picking the low-value bandit using different exploration strategies (a) a high (versus low) value-free random exploration increases the selection of the low-value bandit, whereas neither (b) a high (versus low) novelty exploration, (c) a high (versus low) Thompson-sampling exploration nor (d) a high (versus low) UCB exploration affected this frequency.

**Figure supplement 4.** Simulation illustrations of high and low exploration choice consistency using different exploration strategies shows that (a) a high (versus low) value-free random exploration decreases the proportion of same choices, whereas neither (b) a high (versus low) novelty exploration, (c) a high (versus low) Thompson-sampling exploration nor (d) a high (versus low) UCB exploration affected this measure.

**Figure supplement 5.** Simulation illustrations of high and low exploration on the frequency of picking the novel bandit using different exploration strategies shows that (a) a high (versus low) value-free random exploration has little effect on the selection of the novel bandit, whereas (b) a high (versus low) novelty exploration increases this frequency.

effectively assess all exploration strategies, we opted to present only three of the four different bandit types on each trial, as different bandit triples allow different explorations to manifest. Lastly, to assess whether subjects' behaviour captured exploration, we manipulated the degree to which subjects could interact with the same bandits. Similar to previous studies (*Wilson et al., 2014*), subjects could perform either one draw, encouraging exploitation (short horizon condition), or six draws, encouraging more substantial explorative behaviour (long horizon condition) (*Wilson et al., 2014*; *Warren et al., 2017*).

## Testing the role of catecholamines noradrenaline and dopamine

In a double-blind, placebo-controlled, between-subjects study design, we assigned subjects (N=60) randomly to one of three experimental groups: amisulpride, propranolol, or placebo. The first group received 40 mg of the $\beta$-adrenoceptor antagonist propranolol to alter noradrenaline function, while the second group was administered 400 mg of the D2/D3 antagonist amisulpride that alters dopamine function. Because of different pharmacokinetic properties, these drugs were administered at different times (*Figure 1d*) and compared to a placebo group that received a placebo at both drug times to match the corresponding antagonist's time. One subject (amisulpride group) was excluded from the analysis due to a lack of engagement with the task. Reported findings were corrected for IQ and mood, as drug groups differed marginally in those measures (*Appendix 2—table 7*), by adding WASI (*Wechsler, 2013*) and PANAS (*Watson et al., 1988a*) negative scores as covariates in each ANOVA. Similar results were obtained in an analysis that corrected for physiological effects as from the analysis without covariates (cf. Appendix 1).

## Increased exploration when information can subsequently be exploited

Our task embodied two decision-horizon conditions, a short and a long. To assess whether subjects explored more in a long horizon condition, in which additional information can inform later choices, we examined which bandit subjects chose in their first draw (in accordance with the horizon task [*Wilson et al., 2014*]), irrespective of their drug group. A marker of exploration here is evident if subjects chose bandits with lower expected values, computed as the mean value of their initial samples shown (trials where the novel bandit was chosen were excluded). As expected, subjects chose bandits with a lower expected value in the long compared to the short horizon (repeated-measures ANOVA for the expected value: F(1, 56) = 19.457, p<0.001, $\eta^2$ = 0.258; *Figure 2a*). To confirm that this was a consequence of increased exploration, we analysed the proportion of how often the high-value option was chosen (i.e. the bandit with the highest expected reward based on its initial samples) and we found that subjects (especially those with higher IQ) sampled from it more in the short compared to the long horizon, (WASI-by-horizon interaction: F(1, 54) = 13.304, p = 0.001, $\eta^2$ = 0.198; horizon main effect: F(1, 54) = 3.909, p = 0.053, $\eta^2$ = 0.068; *Figure 3a*), confirming a reduction in exploitation when this information could be subsequently used. Interestingly, this frequency seemed to be marginally higher in the amisulpride group, suggesting an overall higher tendency to exploitation following dopamine blockade (cf. Appendix 1). This horizon-specific behaviour resulted in a lower reward on the first sample in the long compared to the short horizon (F(1, 56) = 23.922, p<0.001, $\eta^2$ = 0.299; *Figure 2c*). When we tested whether subjects were more likely to choose options they knew less about (computed as the mean number of initial samples shown), we found that subjects chose less known (i.e. more informative) bandits more often in the long horizon compared to the short horizon (F(1, 56) = 58.78, p<0.001, $\eta^2$ = 0.512; *Figure 2b*).

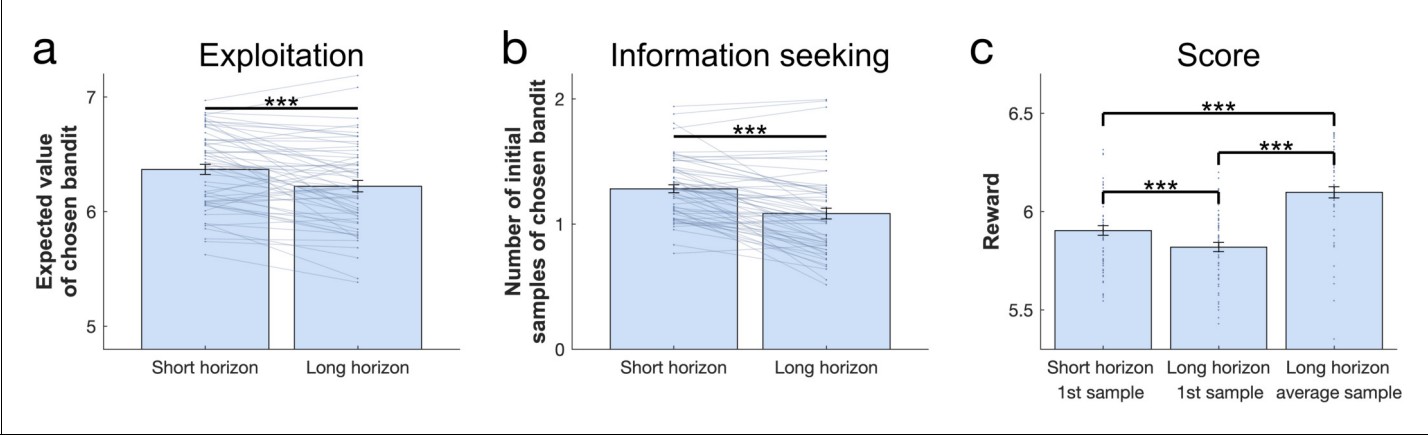

**Figure 2.** Benefits of exploration. To investigate the effect of information on performance we collapsed subjects over all three treatment groups. (**a**) The expected value (average of its initial samples) of the first chosen bandit as a function of horizon. Subjects chose bandits with a lower expected value (i.e. they explored more) in the long horizon compared to the short horizon. (**b**) The mean number of samples for the first chosen bandit as a function of horizon. Subjects chose less known (i.e. more informative) bandits more in the long compared to the short horizon. (**c**) The first draw in the long horizon led to a lower reward than the first draw in the short horizon, indicating that subjects sacrificed larger initial outcomes for the benefit of more information. This additional information helped making better decisions in the long run, leading to a higher earning over all draws in the long horizon. For values and statistics *Appendix 2—table 3*. For response times and details about all long horizons' samples *Figure 2—figure supplement 1*. *** = p<0.001. Data are shown as mean ± 1 SEM and each dot/line represent a subject.

The online version of this article includes the following figure supplement(s) for figure 2:

**Figure supplement 1.** Further analysis of long horizon draws.

Next, to evaluate whether subjects used the additional information beneficially in the long horizon condition, we compared the average reward (across six draws) obtained in the long compared to short horizon (one draw). We found that the average reward was higher in the long horizon (F(1, 56) = 103.759, p<0.001, $\eta^2$ = 0.649; *Figure 2c*), indicating that subjects tended to choose less optimal bandits at first but subsequently learnt to appropriately exploit the harvested information to guide choices of better bandits in the long run. Additionally, when looking specifically at the long horizon condition, we found that subjects earned more when their first draw was explorative versus exploitative (*Figure 2—figure supplement 1c–d*; cf. Appendix 2 for details).

## Subjects demonstrate value-free random behaviour

Value-free random exploration (analogue to $\epsilon$-greedy) predicts that $\epsilon$ % of the time each option will have an equal probability of being chosen. In such a regime (compared to more complex strategies that would favour options with a higher expected value with a similar uncertainty), the probability of choosing bandits with a low expected value (here the low-value bandit; *Figure 1e*) will be higher (*Figure 1—figure supplement 3*). We investigated whether the frequency of picking the low-value bandit was increased in the long horizon condition across all subjects (i.e. when exploration is useful), and we found a significant main effect of horizon (F(1, 54) = 4.069, p = 0.049, $\eta^2$ = 0.07; *Figure 3b*). This demonstrates that value-free random exploration is utilised more when exploration is beneficial.

## Value-free random behaviour is modulated by noradrenaline function

When we tested whether value-free random exploration was sensitive to neuromodulatory influences, we found a difference in how often drug groups sampled from the low-value option (drug main effect: F(2, 54) = 7.003, p = 0.002, $\eta^2$ = 0.206; drug-by-horizon interaction: F(2, 54) = 2.154, p = 0.126, $\eta^2$ = 0.074; *Figure 3b*). This was driven by the propranolol group choosing the low-value option significantly less often than the other two groups (placebo vs propranolol: t(40) = 2.923, p = 0.005, d = 0.654; amisulpride vs propranolol: t(38) = 2.171, p = 0.034, d = 0.496) with no difference between amisulpride and placebo: (t(38) = -0.587, p = 0.559, d = 0.133). These findings demonstrate that a key feature of value-free random exploration, the frequency of choosing low-value bandits, is sensitive to influences from noradrenaline.

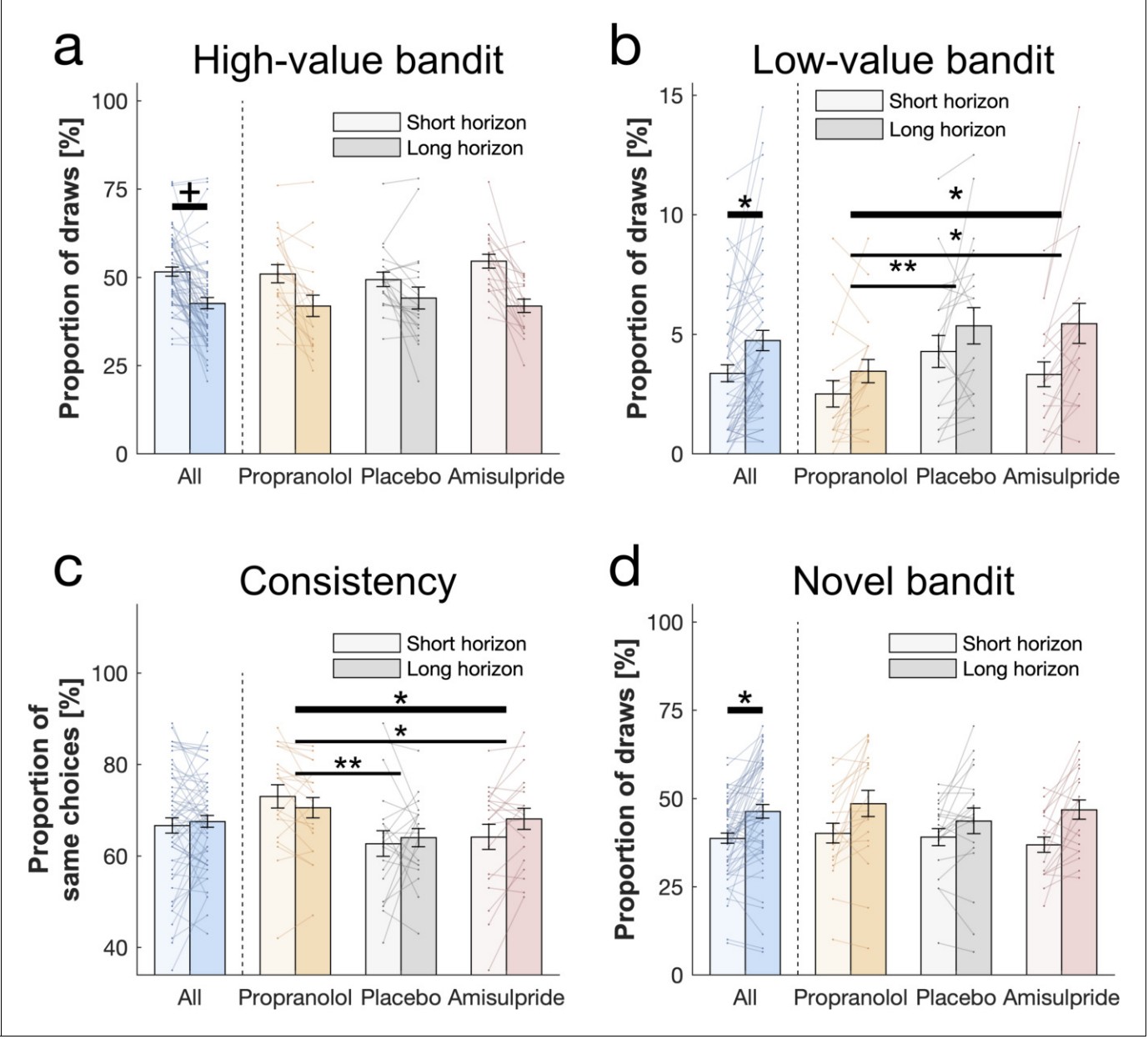

**Figure 3.** Behavioural horizon and drug effects. Choice patterns in the first draw for each horizon and drug group (propranolol, placebo and amisulpride). (a) Subjects sampled from the high-value bandit (i.e. bandit with the highest average reward of initial samples) more in the short horizon compared to the long horizon indicating reduced exploitation. (b) Subjects sampled from the low-value bandit more in the long horizon compared to the short horizon indicating value-free random exploration, but subjects in the propranolol group sampled less from it overall, and (c) were more consistent in their choices overall, indicating that noradrenaline blockade reduces value-free random exploration. (d) Subjects sampled from the novel bandit more in the long horizon compared to the short horizon indicating novelty exploration. Please note that some horizon effects were modulated by subjects' intellectual abilities when additionally controlling for them (*Appendix 2—table 4*). Horizontal bars represent rm-ANOVA (thick) and pairwise comparisons (thin). † = p<0.07, * = p<0.05, ** = p<0.01. Data are shown as mean ± 1 SEM and each line represent one subject. For values and statistics *Appendix 2—table 4*. For response times and frequencies specific to the displayed bandits *Figure 3—figure supplements 1–2*. The online version of this article includes the following figure supplement(s) for figure 3:

**Figure supplement 1.** Response time (RT) analysis per bandit.

**Figure supplement 2.** Proportion of draws per bandit combination (x-axis).

To further examine drug effects on value-free random exploration, we assessed a second prediction, namely choice consistency. Because value-free random exploration ignores all prior information and chooses randomly, it should result in a decreased choice consistency when presented identical choice options (*Figure 1—figure supplements 2* and *4*, compared to more complex strategies which are always biased towards the rewarding or the information providing bandit for example). To this end, each trial was duplicated in our task, allowing us to compute the consistency as the percentage of time subjects sampled from an identical bandit when facing the exact same choice options. In line with the above analysis, we found a difference in consistency by which drug groups sampled from different option (drug main effect: $F(2, 54) = 7.154$, $p = 0.002$, $\eta^2 = 0.209$; horizon main effect: $F(1, 54) = 1.333$, $p = 0.253$, $\eta^2 = 0.024$; drug-by-horizon interaction: $F(2, 54) = 3.352$, $p = 0.042$, $\eta^2 = 0.11$; *Figure 3c*), driven by the fact that the propranolol group chose significantly more consistently than the other two groups (pairwise comparisons: placebo vs propranolol: $t(40) = -3.525$, $p = 0.001$, $d = 0.788$; amisulpride vs placebo: $t(38) = 1.107$, $p = 0.272$, $d = 0.251$; amisulpride vs propranolol: $t(38) = -2.267$, $p = 0.026$, $d = 0.514$). Please see Appendix 1 for further discussion and analysis of the drug-by-horizon interaction. Taken together, these results indicate that value-free random exploration depends critically on noradrenaline functioning, such that an attenuation of noradrenaline leads to a reduction in value-free random exploration.

## Novelty exploration is unaffected by catecholaminergic drugs

Next, we examined whether subjects show evidence for novelty exploration by choosing the novel bandit for which there was no prior information (i.e. no initial samples), as predicted by model simulations (*Figure 1f*). We found a significant main effect of horizon ($F(1, 54) = 5.593$, $p = 0.022$, $\eta^2 = 0.094$; WASI-by-horizon interaction: $F(1, 54) = 13.897$, $p<0.001$, $\eta^2 = 0.205$; *Figure 3d*) indicating that subjects explored the novel bandit significantly more often in the long horizon condition, and this was particularly strong for subjects with a higher IQ. We next assessed whether novelty exploration was sensitive to our drug manipulation, but found no drug effects on the novel bandit ($F(2, 54) = 1.498$, $p = 0.233$, $\eta^2 = 0.053$; drug-by-horizon interaction: $F(2, 54) = 0.542$, $p = 0.584$, $\eta^2 = 0.02$; *Figure 3d*). Thus, there was no evidence that an attenuation of dopamine or noradrenaline function impacts novelty exploration in this task.

## Subjects combine computationally demanding strategies and exploration heuristics

To examine the contributions of different exploration strategies to choice behaviour, we fitted a set of computational models to subjects' behaviour, building on models developed in previous studies (*Gershman, 2018*). In particular, we compared models incorporating UCB, Thompson sampling, an $\epsilon$-greedy algorithm and the novelty bonus (cf. Materials and methods). Essentially, each model makes different exploration predictions. In the Thompson model, Thompson sampling (*Thompson, 1933*; *Agrawal and Goyal, 2012*) leads to an uncertainty-driven value-based random exploration, where both expected value and uncertainty contribute to choice. In this model higher uncertainty leads to more exploration such that instead of selecting a bandit with the highest mean, bandits are chosen relative to how often a random sample would yield the highest outcome, thus accounting for uncertainty (*Schulz and Gershman, 2019*). The UCB model (*Auer, 2003*; *Carpentier et al., 2011*), capturing directed exploration, predicts that each bandit is chosen according to a mixture of expected value and an additional expected information gain (*Schulz and Gershman, 2019*). This is realised by adding a bonus to the expected value of each option, proportional to how informative it would be to select this option (i.e. the higher the uncertainty in the option's value, the higher the information gain). This computation is then passed through a softmax decision model, capturing value-based random exploration. Novelty exploration is a simplified version of the information bonus in the UCB algorithm, which only applies to entirely novel options. It defines the intrinsic value of selecting a bandit about which nothing is known, and thus saves demanding computations of uncertainty for each bandit. Last, the value-free random $\epsilon$-greedy algorithm selects any bandit $\epsilon$ % of the time, irrespective of the prior information of this bandit. For additional models cf. Appendix 1.

We used cross-validation for model selection (*Figure 4a*) by comparing the likelihood of held-out data across different models, an approach that adequately arbitrates between model accuracy and

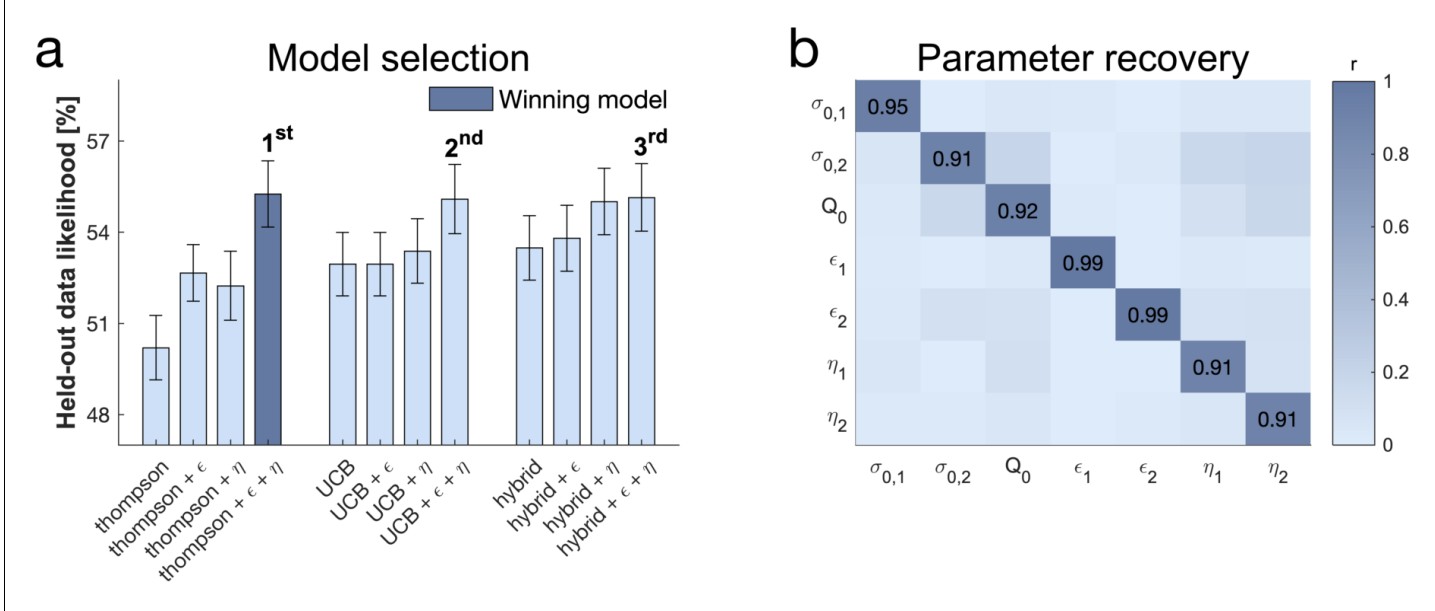

**Figure 4.** Subjects use a mixture of exploration strategies. (**a**) A 10-fold cross-validation of the likelihood of held-out data was used for model selection (chance level = 33.3%; for model selection at the individual level *Figure 4—figure supplement 1*). The Thompson model with both the $\epsilon$-greedy parameter and the novelty bonus $\eta$ best predicted held-out data (**b**) Model simulation with $4^7$ simulations predicted good recoverability of model parameters (for correlations between behaviour and model parameters *Figure 4—figure supplement 2*); $\sigma_0$ is the prior variance and $Q_0$ is the prior mean (for parameter recovery correlation plots *Figure 4—figure supplement 3*). 1 stands for short horizon- and 2 for long horizon-specific parameters. For values and parameter details *Appendix 2—table 5*.

The online version of this article includes the following figure supplement(s) for figure 4:

**Figure supplement 1.** Model comparison: further evaluations.
**Figure supplement 2.** Correlations between model parameters and behaviour.
**Figure supplement 3.** Parameter recovery analysis details.

complexity. The winning model encompasses uncertainty-driven value-based random exploration (Thompson sampling) with value-free random exploration ($\epsilon$-greedy parameter) and novelty exploration (novelty bonus parameter $\eta$). The winning model predicted held-out data with a 55.25% accuracy (SD=8.36%; chance level = 33.33%). Similarly to previous studies (*Gershman, 2018*), the hybrid model combining UCB and Thompson sampling explained the data better than each of those processes alone, but this was no longer the case when accounting for novelty and value-free random exploration (*Figure 4a*). The winning model further revealed that all parameter estimates could be accurately recovered (*Figure 4b*; *Figure 4—figure supplement 3*). Interestingly, although the second and third place models made different prediction about the complex exploration strategy, using a directed exploration with value-based random exploration (UCB) or a combination of complex strategies (hybrid) respectively, they share the characteristic of benefitting from value-free random and novelty exploration. This highlights that subjects used a mixture of computationally demanding and heuristic exploration strategies.

## Noradrenaline controls value-free random exploration

To more formally compare the impact of catecholaminergic drugs on different exploration strategies, we assessed the free parameters of the winning model between drug groups (*Figure 5*, cf. Appendix 2 Table 6 for exact values). First, we examined the $\epsilon$-greedy parameter that captures the contribution of value-free random exploration to choice behaviour. We assessed how this value-free random exploration differed between drug groups. A significant drug main effect (drug main effect: F(2, 54) = 6.722, p = 0.002, $\eta^2$ = 0.199; drug-by-horizon interaction: F(2, 54) = 1.305, p = 0.28, $\eta^2$ = 0.046; *Figure 5a*) demonstrates that the drug groups differ in how strongly they deploy this exploration strategy. Post-hoc analysis revealed that subjects with reduced noradrenaline functioning had the lowest values of $\epsilon$ (pairwise comparisons: placebo vs propranolol: t(40) = 3.177, p = 0.002,

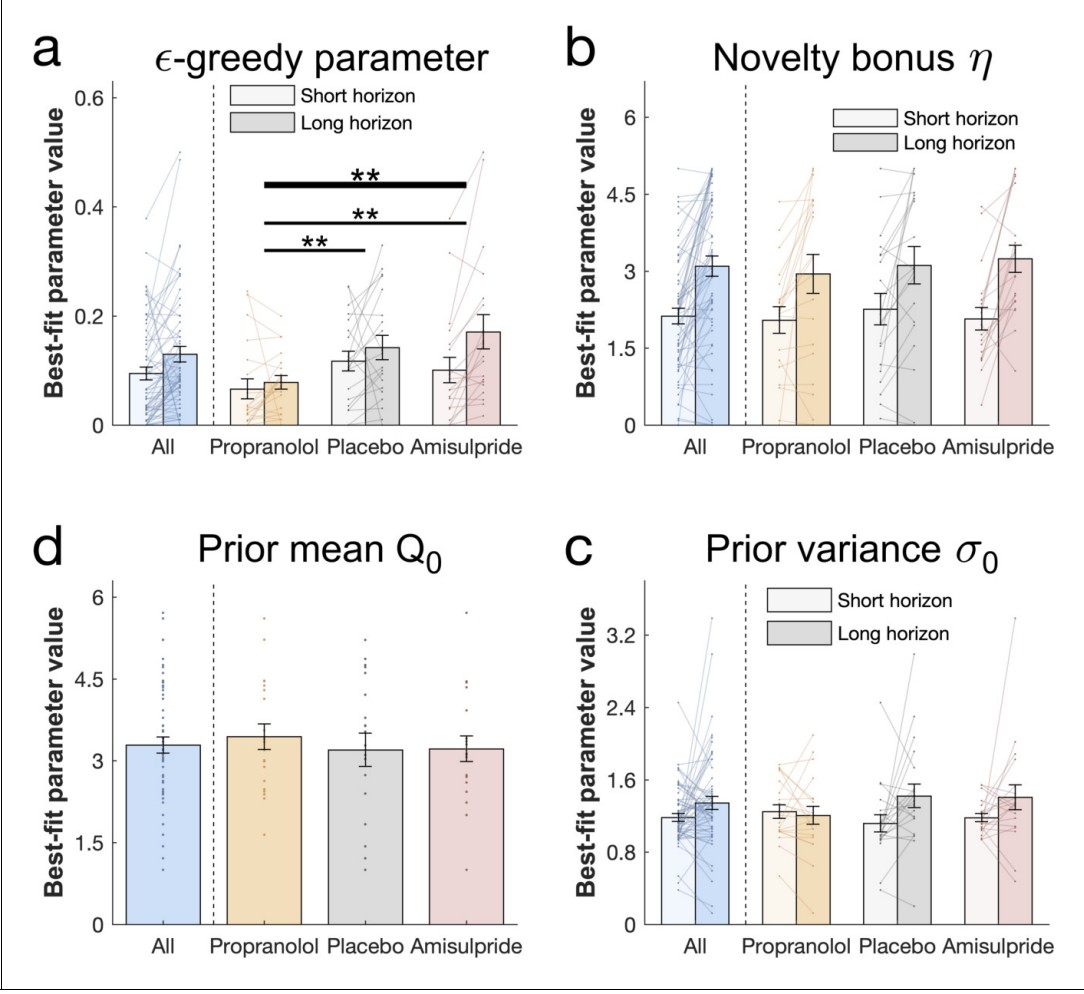

**Figure 5.** Drug effects on model parameters. The winning model's parameters were fitted to each subject's first draw (for model simulations *Figure 5—figure supplement 1*). (a) Subjects had higher values of $\epsilon$ (value-free random exploration) in the long compared to the short horizon. Notably, subjects in the propranolol group had lower values of $\epsilon$ overall, indicating that attenuation of noradrenaline functioning reduces value-free random exploration. Subjects from all groups (b) assigned a similar value to novelty, captured by the novelty bonus $\eta$, which was higher (more novelty exploration) in the long compared to the short horizon. (c) The groups had similar beliefs $Q_0$ about a bandit's mean before seeing any initial samples and (d) were similarly uncertain $\sigma_0$ about it (for gender effects *Figure 5—figure supplement 2*). Please note that some horizon effects were modulated by subjects' intellectual abilities when additionally controlling for them (*Appendix 2—table 6*). ** = p<0.01. Data are shown as mean ± 1 SEM and each dot/line represent one subject. For parameter values and statistics *Appendix 2—table 6*.

The online version of this article includes the following figure supplement(s) for figure 5:

**Figure supplement 1.** Simulated behaviour for Thompson+$\epsilon$+$\eta$ model.

**Figure supplement 2.** Gender effect on prior variance parameter.

**Figure supplement 3.** Simulated behaviour for UCB+$\epsilon$+$\eta$ model.

d = 0.71; amisulpride vs propranolol: t(38) = 2.723, p = 0.009, d = 0 .626) with no significant difference between amisulpride and placebo: (t(38) = 0.251, p = 0.802, d = 0.057). Critically, the effect on $\epsilon$ was also significant when the complex exploration strategy was a directed exploration with value-based random exploration (second place model) and, marginally significant, when it was a combination of the above (third place model; cf. Appendix 1).

The $\epsilon$-greedy parameter was also closely linked to the above behavioural metrics (correlation between the $\epsilon$-greedy parameter and: draws from the low-value bandit: $R_{Pearson}$ = 0.828, p<0.001; choice consistency: $R_{Pearson}$ = -0.596, p<0.001; *Figure 4—figure supplement 2*), and showed a similar horizon effect (horizon main effect: F(1, 54) = 1.968, p = 0.166, $\eta^2$ = 0.035; WASI-by-horizon interaction: F(1, 54) = 6.08, p = 0.017, $\eta^2$ = 0.101; *Figure 5a*). Our findings thus accord with the

model-free analyses and demonstrate that noradrenaline blockade reduces value-free random exploration.

## No drug effects on other parameters

The novelty bonus $\eta$ captures the intrinsic reward of selecting a novel option. In line with the model-free behavioural findings, there was no difference between drug groups in terms of this effect (F(2, 54) = 0.249, p = 0.78, $\eta^2$ = 0.009; drug-by-horizon interaction: F(2, 54) = 0.03, p = 0.971, $\eta^2$ = 0.001). There was also a close alignment between model-based and model-agnostic analyses (correlation between the novelty bonus $\eta$ and draws from the novel bandit: $R_{Pearson}$ = 0.683, p<0.001; *Figure 4—figure supplement 2*), and we found a similarly increased novelty bonus effect in the long horizon in subjects with a higher IQ (WASI-by-horizon interaction: F(1, 54) = 8.416, p = 0.005, $\eta^2$ = 0.135; horizon main effect: F(1, 54) = 1.839, p = 0.181, $\eta^2$ = 0.033; *Figure 5b*).

When analysing the additional model parameters, we found that subjects had similar prior beliefs about bandits, given by the initial estimate of a bandit's mean (prior mean $Q_0$: F(2, 54) = 0.118, p = 0.889, $\eta^2$ = 0.004; *Figure 5c*) and their uncertainty about it (prior variance $\sigma_0$: horizon main effect: F(1, 54) = 0.129, p = 0.721, $\eta^2$ = 0.002; drug main effect: F(2, 54) = 0.06, p = 0.942, $\eta^2$ = 0.002; drug-by-horizon interaction: F(2, 54) = 2.162, p = 0.125, $\eta^2$ = 0.074; WASI-by-horizon interaction: F(1, 54) = 0.022, p = 0.882, $\eta^2$ < 0.001; *Figure 5d*). Interestingly, our dopamine manipulation seemed to affect this uncertainty in a gender-specific manner, with female subjects having larger values of $\sigma_0$ compared to males in the placebo group, and with the opposite being true in the amisulpride group (cf. Appendix 1). Taken together, these findings show that value-free random exploration was most sensitive to our drug manipulations.

## Discussion

Solving the exploration-exploitation problem is non-trivial, and one suggestion is that humans solve it using computationally demanding exploration strategies (*Gershman, 2018*; *Schulz and Gershman, 2019*), taking account of the uncertainty (variance) as well as the expected reward (mean) of each choice. Although tracking the distribution of summary statistics (e.g. mean and variance) is less resource costly than keeping track of full distributions (*D'Acremont and Bossaerts, 2008*), it nevertheless carries considerable costs when one has to keep track of multiple options, as in exploration. Indeed, in a three-bandit task such as that considered here, this results in a necessity to compute six key-statistics, drastically limiting computational resources when selecting among choice options (*Cogliati Dezza et al., 2019*). Real-life decisions often comprise an unlimited range of options, which results in a tracking of a multitude of key-statistics, potentially mandating a deployment of alternative more efficient strategies. Here, we demonstrate that two additional, less resource-hungry heuristics are at play during human decision-making, value-free random exploration and novelty exploration.

By assigning intrinsic value (novelty bonus [*Krebs et al., 2009*]) to an option not encountered before (*Foley et al., 2014*), a novelty bonus can be seen as an efficient simplification of demanding algorithms, such as UCB (*Auer, 2003*; *Carpentier et al., 2011*). It is interesting to note that our winning model did not include UCB, but instead novelty exploration. This indicates humans might use such a novelty shortcut to explore unseen, or rarely visited, states to conserve computational costs when such a strategy is possible. A second exploration heuristic that also requires minimal computational resources, value-free random exploration, also plays a role in our task. Even though less optimal, its simplicity and neural plausibility renders it a viable strategy. Indeed, we observe an increase in performance in each model after adding $\epsilon$, supporting the notion that this strategy is a relevant additional human exploration heuristic. Interestingly, the benefit of $\epsilon$ is somewhat smaller in a simple UCB model (without novelty bonus), which probably arises because value-based random exploration partially captures some of the increased noisiness. We show through converging behavioural and modelling measures that both value-free random and novelty exploration were deployed in a goal-directed manner, coupled with increased levels of exploration when this was strategically useful. Importantly, these heuristics were observed in all best models (first, second and third position) even though each incorporated different exploration strategies. This suggests that the complex models make similar predictions in our task. This is also observed in our simulations, and demonstrates that

value-free random exploration is at play even when accounting for other value-based forms of random exploration (*Gershman, 2018*; *Wilson et al., 2014*), whether fixed or uncertainty-driven.

Exploration was captured in a similar manner to previous studies (*Wilson et al., 2014*), by comparing in the same setting (i.e. same prior information) the first choice in a long decision horizon, where reward can be increased in the long term through information gain, and in a short decision horizon where information cannot subsequently be put to use. This means that by changing the opportunity to benefit from the information gained for the first sample, the long horizon invites extended exploration (*Wilson et al., 2014*), what we find also in our study. This experimental manipulation is a well-established means for altering exploration and has been used extensively in previous studies (*Wilson et al., 2014*; *Zajkowski et al., 2017*; *Warren et al., 2017*; *Wu et al., 2018*). Nevertheless, there remains a possibility that a longer horizon may also affect the psychological nature of the task. In our task, reward outcomes were presented immediately after every draw, rendering it unlikely that perception of reward delays (i.e. delay discounting) is impacted. Moreover, a monetary bonus was given only at the end of the task, and thus did not impact the horizon manipulation. We also consider our manipulation was unlikely to change effort in each horizon, because the reward (i.e. size of the apple) remains the same at every draw, resulting in an equivalent reward-effort ratio (*Skvortsova et al., 2014*; *Hauser et al., 2017a*; *Walton and Bouret, 2019*; *Salamone et al., 2016*). However, this issue can be addressed in further studies, for example, by equating the amount of button presses across both conditions.

Value-free random exploration might reflect other influences, such as attentional lapses or impulsive motor responses. We consider these as unlikely to a significant factor at play here. Indeed, there are two key features that would signify such effects. Firstly, these influences would be independent of task condition. Secondly, they would be expected to lead to shorter, or more variable, response latencies. In our data, we observe an increase in value-free exploration in the long horizon condition in both behavioural measures and model parameters, speaking against an explanation based upon simple mistakes. Moreover, we did not observe a difference in response latency for choices that were related to value-free random exploration (cf. Appendix 1), further arguing against mistakes. Lastly, the sensitivity of value-free random exploration to propranolol supports this being a separate process, and previous studies using the same drug did not find an effect on task mistakes (e.g. on accuracy [*Hauser et al., 2018*; *Jahn et al., 2018*; *Salamone et al., 2016*; *Hauser et al., 2019*; ; *Sokol-Hessner et al., 2015*]). However, future studies could explore these exploration strategies in more detail including by reference to subjects' own self-reports.

It is still unclear how exploration strategies are implemented neurobiologically. Noradrenaline inputs, arising from the locus coeruleus (LC; *Rajkowski et al., 1994*) are thought to modulate exploration (*Schulz and Gershman, 2019*; *Aston-Jones and Cohen, 2005*; *Servan-Schreiber et al., 1990*), although empirical data on its precise mechanisms and means of action remains limited. In this study, we found that noradrenaline impacted value-free random exploration, in contrast to novelty exploration and complex exploration. This might suggest that noradrenaline influences ongoing valuation or choice processes that discards prior information. Importantly, this effect was observed whether the complex exploration was an uncertainty-driven value-based random exploration (winning model), a directed exploration with value-based random exploration (second place model) or a combination of the above (third place model; cf. Appendix 1). This is consistent with findings in rodents where enhanced anterior cingulate noradrenaline release leads to more random behaviour (*Tervo et al., 2014*). It is also consistent with pharmacological findings in monkeys that show enhanced choice consistency after reducing LC noradrenaline firing rates (*Jahn et al., 2018*). It would be interesting for future studies to determine, in more detail, whether value-free random exploration is corrupting a value computation itself, or whether it exclusively biases the choice process.

We note that pupil diameter has been used as an indirect marker of noradrenaline activity (*Joshi et al., 2016*), although the link between the two it not always straightforward (*Hauser et al., 2019*). Because the effect of pharmacologically induced changes of noradrenaline levels on pupil size remains poorly understood (*Hauser et al., 2019*; *Joshi and Gold, 2020*), including the fact that previous studies found no effect of propranolol on pupil diameter (*Hauser et al., 2019*; *Koudas et al., 2009*), we opted against using pupillometry in this study. However, our current findings align with previous human studies that show an association between this indirect marker and exploration, but that study did not dissociate between the different potential exploration strategies

that subjects could deploy (*Jepma and Nieuwenhuis, 2011*). Future studies might usefully include indirect measures of noradrenaline activity, for example pupillometry, to examine a potential link between natural variations in noradrenaline levels and a propensity towards value-free random exploration.

The LC has two known modes of synaptic signalling (*Rajkowski et al., 1994*), tonic and phasic, thought to have complementary roles (*Dayan and Yu, 2006*). Phasic noradrenaline is thought to act as a reset button (*Dayan and Yu, 2006*), rendering an agent agnostic to all previously accumulated information, a de facto signature of value-free random exploration. Tonic noradrenaline has been associated, although not consistently (*Jepma et al., 2010*), with increased exploration (*Aston-Jones and Cohen, 2005*; *Usher et al., 1999*), decision noise in rats (*Kane et al., 2017*) and more specifically with random as opposed to directed exploration strategies (*Warren et al., 2017*). This later study unexpectedly found that boosting noradrenaline decreased (rather than increased) random exploration, which the authors speculated was due to an interplay with phasic signalling. Importantly, the drug used in that study also affects dopamine function making it difficult to assign a precise interpretation to the finding. A consideration of this study influenced our decision to opt for drugs with high specificity for either dopamine or noradrenaline (*Hauser et al., 2018*), enabling us to reveal highly specific effects on value-free random exploration. Although the contributions of tonic and phasic noradrenaline signalling cannot be disentangled in our study, our findings align with theoretical accounts and non-primate animal findings, indicating that phasic noradrenaline promotes value-free random exploration.

Aside from this 'reset signal' role, noradrenaline has been assigned other roles, including a role in memory function (*Sara et al., 1994*; *Rossetti and Carboni, 2005*; *Gibbs et al., 2010*). To minimise a possible memory-related impact, we designed the task such that all necessary information was visible on the screen at all times. This means subjects did not have to memorise values for a given trial, rendering the task less susceptible to forgetting or other memory effects. Another role for noradrenaline relates to volatility and uncertainty estimation (*Silvetti et al., 2013*; *Yu and Dayan, 2005*; *Nassar et al., 2012*), as well as the energisation of behaviour (*Varazzani et al., 2015*; *Silvetti et al., 2018*). Non-human primate studies demonstrate a higher LC activation for high effort choices, suggesting that noradrenaline release facilitates energy mobilisation (*Varazzani et al., 2015*). Theoretical models also suggest that the LC is involved in the control of effort exertion. Thus, it is thought to contribute to trading off between effortful actions leading to large rewards and 'effortless' actions leading to small rewards by modulating 'raw' reward values as a function of the required effort (*Silvetti et al., 2018*). Our task can be interpreted as encapsulating such a trade-off: complex exploration strategies are effortful but optimal in terms of reward gain, while value-free random exploration requires little effort while occasionally leading to low reward. Applying this model, a noradrenaline boost could optimise cognitive effort allocation for high reward gain (*Silvetti et al., 2018*), thereby facilitating complex exploration strategies compared to value-free random exploration. In such a framework, blocking noradrenaline release should decrease usage of complex exploration strategies, leading to an increase of value-free random exploration which is the opposite of what we observed in our data. Another interpretation of an effort-facilitation model of noradrenaline is that a boost would help overcoming cost, that is the lack of immediate reward when selecting the low-value bandit, essentially providing a significant increase to the value of information gain. In line with our results, a decrease would interrupt this boost in valuation, removing an incentive to choose the low-value option. However, this theory is currently limited by the absence of empirical evidence for noradrenaline boosting valuation.

Noradrenaline blockade by propranolol has been shown previously to enhance metacognition (*Hauser et al., 2017b*), decrease information gathering (*Hauser et al., 2018*), and attenuate arousal-induced boosts in incidental memory (*Hauser et al., 2019*). All these findings, including a decrease in value-free random exploration found here, suggests propranolol may influence how neural noise affects information processing. In particular, the results indicate that under propranolol behaviour is less stochastic and less influenced by 'task-irrelevant' distractions. This aligns with theoretical ideas, as well as recent optogenetic evidence (*Tervo et al., 2014*), that proposes noradrenaline infuses noise in a temporally targeted way (*Dayan and Yu, 2006*). It also accords with studies implicating noradrenaline in attention shifts (for a review *Trofimova and Robbins, 2016*). Other gain-modulation theories of noradrenaline/catecholamine function have proposed an effect on stochasticity (*Aston-Jones and Cohen, 2005*; *Servan-Schreiber et al., 1990*), although a hypothesised direction of effect

is different (i.e. noradrenaline decreases stochasticity). Several aspects of noradrenaline functioning may explain the contradictory accounts of its link with stochasticity. For example, they might be capturing different aspects of an assumed U-shaped noradrenaline functioning curve, and/or distinct activity modes of noradrenaline (i.e. tonic and phasic firing) (*Aston-Jones and Cohen, 2005*). Further studies can shed light on how different modes of activity affect value-free random exploration. This idea can be extended also to tasks where propranolol has been shown to attenuate a discrimination between different levels of loss (with no effect on the value-based exploration parameter, referred to in these studies as consistency) (*Rogers et al., 2004*) and a reduction in loss aversion (*Sokol-Hessner et al., 2015*). This hints at additional roles for noradrenaline on prior information and task-distractibility during exploration in loss-frame environments. Future studies investigating exploration in loss contexts might provide important additional information on these questions.

It is important to mention here that β-adrenergic receptors, the primary target of propranolol, have been shown (unlike $\alpha$-adrenergic receptors) to increase synaptic inhibition within rat cortex (*Waterhouse et al., 1982*), specifically through inhibitory GABA-mediated transmission (*Waterhouse et al., 1984*). Additionally β-adrenergic receptors are more concentrated in the intermediate layers in the prefrontal area (*Goldman-Rakic et al., 1990*), within which inhibition is favoured (*Isaacson and Scanziani, 2011*). Thus, inhibitory mechanisms might account for noradrenaline-related task-distractibility and randomness, or the role of β-adrenergic receptors in executive function impairments (*Salgado et al., 2016*). This raises the question of whether blocking β-adrenergic receptors might lead to an accumulation of synaptic noradrenaline, and therefore act via $\alpha$-adrenergic receptors. To the best of our knowledge, evidence for such an effect is limited. A second question is whether the observed effects are a pure consequence of propranolol's impact on the brain, or whether they reflect peripheral effects of propranolol. When we examined peripheral markers (i.e. heart rate) and behaviour we found no evidence for an effect on any of our findings, rendering such influences unlikely. However, future studies using drugs that exclusively targets peripheral, but not central, noradrenaline receptors (e.g. *De Martino et al., 2008*) are needed to answer this question conclusively.

Dopamine has been ascribed multiple functions besides reward learning (*Schultz et al., 1997*), such as novelty seeking (*Düzel et al., 2010*; *Wittmann et al., 2008*; *Costa et al., 2014*) or exploration in general (*Frank et al., 2009*). In fact, studies have demonstrated that there are different types of dopaminergic neurons in the ventral tegmental area, and that some contribute to non-reward signals, such as saliency and novelty (*Bromberg-Martin et al., 2010*). This suggests a role in novelty exploration. Moreover, dopamine has been suggested as important in an exploration-exploitation arbitration (*Zajkowski et al., 2017*; *Kayser et al., 2015*; *Chakroun et al., 2019*), although its precise role remains unclear, given reported effects on random exploration (*Cinotti et al., 2019*), on directed exploration (*Costa et al., 2014*; *Frank et al., 2009*), or no effects at all (*Krugel et al., 2009*). A recent study found no effect following dopamine blockade using haloperidol (*Chakroun et al., 2019*), which interestingly also affects noradrenaline function (e.g. *Fang and Yu, 1995*; *Toru and Takashima, 1985*). Our results did not demonstrate any main effect of dopamine manipulation on exploration strategies, even though blocking dopamine was associated with a trend level increase in exploitation (cf. Appendix 1). We believe it unlikely this reflects an ineffective drug dose as previous studies have found neurocognitive effects with the same dose (*Hauser et al., 2019*; *Hauser et al., 2018*; *Kahnt et al., 2015*; *Kahnt and Tobler, 2017*).

One possible reason for an absence of significant findings is that our dopaminergic blockade targets D2/D3 receptors rather than D1 receptors, a limitation due a lack of available specific D1 receptor blockers for use in humans. An expectation of greater D1 involvement arises out of theoretical models (*Humphries et al., 2012*) and a prefrontal hypothesis of exploration (*Frank et al., 2009*). Interestingly, we observed a weak gender-specific differential drug effect on subjects' uncertainty about an expected reward, with women being more uncertain than men in the placebo setting, but more certain in the dopamine blockade setting (cf. Appendix 1). This might be meaningful as other studies using the same drug have also found behavioural gender-specific drug effects (*Soutschek et al., 2017*). Upcoming, novel drugs (*Soutschek et al., 2020*) might be able help unravel a D1 contribution to different forms of exploration. Additionally, future studies could use approved D2/D3 agonists (e.g. ropinirole) in a similar design to probe further whether enhancing dopamine leads to a general increase in exploration.

In conclusion, humans supplement computationally expensive exploration strategies with less resource demanding exploration heuristics, and as shown here the latter include value-free random and novelty exploration. Our finding that noradrenaline specifically influences value-free random exploration demonstrates that distinct exploration strategies may be under specific neuromodulator influence. Our current findings may also be relevant to enabling a richer understanding of disorders of exploration, such as attention-deficit/hyperactivity disorder (*Hauser et al., 2016*; *Hauser et al., 2014*) including how aberrant catecholamine function might contribute to its core behavioural impairments.

## Materials and methods

### Subjects

Sixty healthy volunteers aged 18–35 (mean = 23.22, SD = 3.615) participated in a double-blind, placebo-controlled, between-subjects study. The sample size was determined using power calculations taking effect sizes from our prior studies that used the same drug manipulations (*Hauser et al., 2019*; *Hauser et al., 2018*; *Hauser et al., 2017b*). Each subject was randomly allocated to one of three drug groups, controlling for an equal gender balance across all groups (cf. Appendix 1). Candidate subjects with a history of neurological or psychiatric disorders, current health issues, regular medications (except contraceptives), or prior allergic reactions to drugs were excluded from the study. Subjects had (self-reported) normal or corrected-to-normal vision. The groups consisted of 20 subjects each matched (*Appendix 2—table 1*) for gender and age. To evaluate peripheral drug effects, heart rate, systolic and diastolic blood pressure were collected at three different time-points: 'at arrival', 'pre-task' and 'post-task', cf. Appendix 1 for details. At 50 min after administrating the second drug, subjects were filled in the PANAS questionnaires (*Watson et al., 1988a*) and completed the WASI Matrix Reasoning subtest (*Wechsler, 2013*). Groups differed in mood (PANAS negative affect, cf. Appendix 1 for details) and marginally in intellectual abilities (WASI), and so we control for these potential confounders in our analyses (cf. Appendix 1 for uncorrected results). Subjects were reimbursed for their participation on an hourly basis and received a bonus according to their performance (proportional to the sum of all the collected apples' sizes). One subject from the amisulpride group was excluded due to not engaging in the task and performing at chance level. The study was approved by the UCL research ethics committee and all subjects provided written informed consent.

### Pharmacological manipulation

To reduce noradrenaline functioning, we administered 40 mg of the non-selective β-adrenoceptor antagonist propranolol 60 min before the task (*Figure 1D*). To reduce dopamine functioning, we administered 400 mg of the selective D2/D3 antagonist amisulpride 90 min before the task. Because of different pharmacokinetic properties, drugs were administered at different times. Each drug group received the drug on its corresponding time point and a placebo at the other time point. The placebo group received placebo at both time points, in line with our previous studies (*Hauser et al., 2019*; *Hauser et al., 2018*; *Hauser et al., 2017b*).

### Experimental paradigm

To quantify different exploration strategies, we developed a multi-armed bandit task implemented using Cogent (http://www.vislab.ucl.ac.uk/cogent.php) for MATLAB (R2018a). Subjects had to choose between bandits (i.e. trees) that produced samples (i.e. apples) with varying reward (i.e. size) in two different horizon conditions (*Figure 1a–b*). Bandits were displayed during the entire duration of a trial and there was no time limit for sampling from (choosing) the bandits. The sizes of apples they collected were summed and converted to an amount of juice (feedback), which was displayed during 2000 ms at the end of each trial. Subjects were instructed to endeavour to make the most juice and that they would receive a cash bonus proportional to their performance. Overall subjects received £10/hr and a mean bonus of £1.12 (std: £0.06).

Similar to the horizon task (*Wilson et al., 2014*), to induce different extents of exploration, we manipulated the horizon (i.e. number of apples to be picked: one in the short horizon, six in the long horizon) between trials. This horizon-manipulation, which has been extensively used to modulate

exploratory behaviour (*Zajkowski et al., 2017*; *Warren et al., 2017*; *Wu et al., 2018*; *Guo and Yu, 2018*), promotes exploration in the long horizon condition as there are more opportunities to gather reward.

Within a single trial, each bandit had a different mean reward μ (i.e. apple size) and associated uncertainty as captured by the number of initial samples (i.e. number of apples shown at the beginning of the trial). Each bandit (i.e. tree) $i$ was from one of four generative processes (*Figure 1c*) characterised by different means $\mu_i$ and number of initial samples. The rewards (apple sizes) for each bandit were sampled from a normal distribution with mean $\mu_i$, specific to the bandit, and with a fixed variance, $S^2 = 0.8$. The rewards were those sampled values rounded to the closest integer. Each distribution was truncated to [2, 10], meaning that rewards with values above or below this interval were excluded, resulting in a total of 9 possible rewards (i.e. 9 different apple sizes; *Figure 1—figure supplement 1* for a representation). The 'certain standard bandit' provided three initial samples and on every trial its mean $\mu_{cs}$ was sampled from a normal distribution: $\mu_{cs} \sim N(5.5, 1.4)$. The 'standard bandit' provided one initial sample and to make sure that its mean $\mu_s$ was comparable to $\mu_{cs}$, the trials were split equally between the four following: $\{\mu_s = \mu_{cs} + 1; \ \mu_s = \mu_{cs} - 1; \ \mu_s = \mu_{cs} + 2; \ \mu_s = \mu_{cs} - 2\}$. The 'novel bandit' provided no initial samples and its mean $\mu_n$ was comparable to both $\mu_{cs}$ and $\mu_s$ by splitting the trials equally between the eight following:

$$\{\mu_n = \mu_{cs} + 1; \ \mu_n = \mu_{cs} - 1; \ \mu_n = \mu_{cs} + 2; \ \mu_n = \mu_{cs} - 2; \ \mu_n = \mu_s + 1; \mu_n = \mu_s - 1; \ \mu_n = \mu_s + 2; \ \mu_n = \mu_s - 2\}.$$

The 'low bandit' provided one initial sample which was smaller than all the other bandits' means on that trial: $\mu_l = min(\mu_{cs}, \mu_s, \mu_n) - 1$. We ensured that the initial sample from the low-value bandit was the smallest by resampling from each bandit in the trials were that was not the case. To make sure that our task captures heuristic exploration strategies, we simulated behaviour (*Figure 1*). Additionally, in each trial, to avoid that some exploration strategies overshadow other ones, only three of the four different groups were available to choose from. Based on the mean of the initial samples, we identified the high-value option (i.e. the bandit with the highest expected reward) in trials where both the certain-standard and the standard bandit were present.

There were 25 trials of each of the four three-bandit combination making it a total of 100 different trials. They were then duplicated to measure choice consistency, defined as the frequency of making the same choice on identical trials (in contrast to a previous propranolol study where consistency was defined in terms of a value-based exploration parameter [*Sokol-Hessner et al., 2015*]). Each subject played these 200 trials both in a short and in a long horizon settings, resulting in a total of 400 trials. The trials were randomly assigned to one of four blocks and subjects were given a short break at the end of each of them. To prevent learning, the bandits' positions (left, middle or right) as well as their colour (eight sets of three different colours) where shuffled between trials. To ensure subjects distinguished different apple sizes and understood that apples from the same tree were always of similar size (generated following a normal distribution), they needed to undergo training prior to the main experiment. In training, based on three displayed apples of similar size, they were tasked to guess between two options, namely which apple was most likely to come from the same tree and then received feedback about their choice.

## Statistical analyses

All statistical analyses were performed using the R Statistical Software (*R Development Core Team, 2011*). For computing ANOVA tests and pairwise comparisons the 'rstatix' package was used, and for computing effect sizes the 'lsr' package (*Navarro, 2015*) was used. To ensure consistent performance across all subjects, we excluded one outlier subject (belonging to the amisulpride group) from our analysis due to not engaging in the task and performing at chance level (defined as randomly sampling from one out of three bandits, that is 33%). Each bandit's selection frequency for a horizon condition was computed over all 200 trials and not only over the trials where this specific bandit was present (i.e. 3/4 of 200 = 150 trials). In all the analysis comparing horizon conditions, except when looking at score values (*Figure 2c*), only the first draw of the long horizon was used. We compared behavioural measures and model parameters using (paired-samples) t-tests and repeated-measures (rm-) ANOVAs with a between-subject factor of drug group (propranolol group, amisulpride group, placebo group) and a within-subject factor horizon (long, short). Information

seeking, expected values and scores were analysed using rm-ANOVAs with a within-subject factor horizon. Measures that were horizon-independent (e.g. prior mean), were analysed using one-way ANOVAs with a between-subject factor drug group. As drug groups differed in negative affect (*Appendix 2—table 1*), which, through its relationship to anxiety (*Watson et al., 1988b*) is thought to affect cognition (*Bishop and Gagne, 2018*) and potentially exploration (*de Visser et al., 2010*). We corrected for negative affect (PANAS) and IQ (WASI) in each analysis by adding those two measures as covariates in each ANOVA mentioned above (cf. Appendix 1 for analysis without covariates and analysis with physiological effect as an additional covariates). We report effect sizes using partial eta squared ($\eta^2$) for ANOVAs and Cohen's d (d) for t-tests (*Richardson, 2011*).

## Computational modelling

We adapted a set of Bayesian generative models from previous studies (*Gershman, 2018*), where each model assumed that different characteristics account for subjects' behaviour. The binary indicators $(c_{tr}, c_n)$ indicate which components (value-free random and novelty exploration respectively) were included in the different models. The value of each bandit is represented as a distribution $N(Q, S)$ with $S = 0.8$, the sampling variance fixed to its generative value. Subjects have prior beliefs about bandits' values which we assume to be Gaussian with mean $Q_0$ and uncertainty $\sigma_0$. The subject's initial estimate of a bandit's mean ($Q_0$; prior mean) and its uncertainty about it ($\sigma_0$; prior variance) are free parameters.

These beliefs are updated according to Bayes rule (detailed below) for each initial sample (note that there are no updates for the novel bandit).

## Mean and variance update rules

At each time point $t$, in which a sample $m$, of one of the bandits is presented, the expected mean $Q$ and precision $\tau = \frac{1}{\sigma^2}$ of the corresponding bandit $i$ are updated as follows:

$$Q_{i,t+1} = \frac{\tau_{i,t} * Q_{i,t} + \tau_{samp} * m}{\tau_{i,t} + \tau_{samp}}$$

$$\tau_{t+1}^i = \tau_{samp} + \tau_t^i$$

where $\tau_{samp} = \frac{1}{S^2}$ is the sampling precision, with the sampling variance $S = 0.8$ fixed. Those update rules are equivalent to using a Kalman filter (*Bishop, 2006*) in stationary bandits.

We examined three base models: the UCB model, the Thompson model, and the hybrid model. The UCB model encompasses the UCB algorithm (captures directed exploration) and a softmax choice function (captures a value-based random exploration). The Thompson model reflects Thompson sampling (captures an uncertainty-driven value-based random exploration). The hybrid model captures the contribution of the UCB model and the Thompson model, essentially a mixture of the above. We computed three extensions of each model by either adding value-free random exploration $(c_{vf}, c_n) = (1, 0)$, novelty exploration $(c_{vf}, c_n) = (0, 1)$ or both heuristics $(c_{vf}, c_n) = (1, 1)$, leading to a total of 12 models (see the labels on the x-axis in *Figure 4a*; $(c_{vf}, c_n) = (0, 0)$ is the model with no extension). For additional models cf. Appendix 1. A coefficient $c_{vf}=1$ indicates that an $\epsilon$-greedy component was added to the decision rule, ensuring that once in a while (every $\epsilon$ % of the time), another option than the predicted one is selected. A coefficient $c_n=1$ indicates that the novelty bonus $\eta$ is added to the computation of the value of novel bandits and the Kronecker delta δ in front of this bonus ensures that it is only applied to the novel bandit. The models and their free parameters (summarised in *Appendix 2—table 5*) are described in detail below.

## Choice rules
### UCB model

In this model, an information bonus $\gamma$ is added to the expected reward of each option, scaling with the option's uncertainty (UCB). The value of each bandit $i$ at timepoint $t$ is:

$$V_{i,t} = Q_{i,t} + \gamma\sigma_{i,t} + c_n\eta\delta_{[i=novel]}$$

The probability of choosing bandit $i$ was given by passing this into the softmax decision function:

$$P(c_t = i) = \frac{e^{\beta V_{i,t}}}{\sum_x e^{\beta V_{i,t}}} * (1 - c_{vf}\epsilon) + c_{vf}\frac{\epsilon}{3}$$

where $\beta$ is the inverse temperature of the softmax (lower values producing more value-based random exploration), and the coefficient $c_{vf}$ adds the value-free random exploration component.

## Thompson model

In this model, based on Thompson sampling, the overall uncertainty can be seen as a more refined version of a decision temperature (**Gershman, 2018**). The value of each band it $i$ is as before:

$$V_{i,t} = Q_{i,t} + c_n\eta\delta_{[i=novel]}$$

A sample $x_{i,t} \sim N\left(V_{i,t}, \sigma_{i,t}^2\right)$ is taken from each bandit. The probability of choosing a bandit $i$ depends on the probability that all pairwise differences between the sample from bandit $i$ and the other bandits $j \neq i$ were greater or equal to 0 (see the probability of maximum utility choice rule [**Speekenbrink and Konstantinidis, 2015**]). In our task, because three bandits were present, two pairwise differences scores (contained in the two-dimensional vector u) were computed for each bandit. The probability of choosing bandit $i$ is:

$$P(c_t = i) = P\left(\forall j: x_{i,t} >; x_{j,t}\right) * (1 - c_{vf}\epsilon) + c_{vf}\frac{\epsilon}{3}$$

$$P(c_t = x_i) = \int_0^\infty \int_0^\infty \Phi\left(u; M_{i,t}, C_{i,t}\right) du * (1 - c_{vf}\epsilon) + c_{vf}\frac{\epsilon}{3}$$

where $\Phi$ is the multivariate Normal density function with mean vector.

$M_{i,t} = A_i \begin{pmatrix} V_{1,t} \\ V_{2,t} \\ V_{3,t} \end{pmatrix}$ and covariance matrix

$$C_{i,t} = A_i \begin{pmatrix} \sigma_{1,t} & 0 & 0 \\ 0 & \sigma_{2,t} & 0 \\ 0 & 0 & \sigma_{3,t} \end{pmatrix} A_i^T$$

Where the matrix $A_i$ computes the pairwise differences between bandit $i$ and the other bandits. For example, for band it $i = 1$:

$$A_1 = \begin{pmatrix} 1 & -1 & 0 \\ 1 & 0 & -1 \end{pmatrix}$$

## Hybrid model

This model allows a combination of the UCB model and the Thompson model. The probability of choosing bandit $i$ is:

$$P(c_t = i) = \left(wP_{UCB}(c_t = i) + (1-w)P_{Thompson}(c_t = i)\right) * (1 - c_{vf}\epsilon) + c_{vf}\frac{\epsilon}{3}$$

where $w$ specifies the contribution of each of the two models. $P_{UCB}$ and $P_{Thompson}$ are calculated for $c_{vf}$=0. If $w$=1, only the UCB model is used while if $w$=0 only the Thompson model is used. In between values indicate a mixture of the two models.

All the parameters besides $Q_0$ and $w$ were free to vary as a function of the horizon (**Appendix 2—table 5**) as they capture different exploration forms: directed exploration (information bonus $\gamma$; UCB model), novelty exploration (novelty bonus $\eta$), value-based random exploration (inverse temperature $\beta$; UCB model), uncertainty-directed exploration (prior variance $\sigma_0$; Thompson model), and value-free random exploration ($\epsilon$-greedy parameter). The prior mean $Q_0$ was fitted to both horizons together as we do not expect the belief of how good a bandit is to depend on the horizon. The same was done for $w$ as assume the arbitration between the UCB model and the Thompson model does not depend on horizon.

## Parameter estimation

To fit the parameter values, we used the maximum a posteriori probability (MAP) estimate. The optimisation function used was fmincon in MATLAB. The parameters could vary within the following bounds: $\sigma_0 = [0.01, 6]$, $Q_0 = [1, 10]$, $\epsilon = [0, 0.5]$, $\eta = [0, 5]$. The prior distribution used for the prior mean parameter $Q_0$ was the normal distribution: $Q_0 \sim N(5, 2)$ that approximates the generative distributions. For the $\epsilon$-greedy parameter, the novelty bonus $\eta$ and the prior variance parameter $\sigma_0$, a uniform distribution (of range equal to the specific parameters' bounds) was used, which is equivalent to performing MLE. A summary of the parameter values per group and per horizon can be found in *Appendix 2—table 6*.

## Model comparison

We performed a K-fold cross-validation with $K = 10$. We partitioned the data of each subject ($N_{trials} = 400$; 200 in each horizon) into K folds (i.e. subsamples). For model fitting in our model selection, we used maximum likelihood estimation (MLE), where we maximised the likelihood for each subject individually (fmincon was ran with eight randomly chosen starting point to overcome potential local minima). We fitted the model using K-1 folds and validated the model on the remaining fold. We repeated this process K times, so that each of the K fold is used as a validation set once, and averaged the likelihood over held out trials. We did this for each model and each subject and averaged across subjects. The model with the highest likelihood of held-out data (the winning model) was the Thompson sampling with $(c_{tr}, c_n) = \{1, 1\}$. It was also the model which accounted best for the largest number of subjects (*Figure 4—figure supplement 1*).

## Parameter recovery

To make sure that the parameters are interpretable, we performed a parameter recovery analysis. For each parameter, we took four values, equally spread, within a reasonable parameter range ($\sigma_0 = [0.5, 2.5]$, $Q_0 = [1, 6]$, $\epsilon = [0, 0.5]$, $\eta = [0, 5]$). All parameters but $Q_0$ were free to vary as a function of the horizon. We simulated behaviour with one artificial agent for each $4^7$ combinations using a new trial for each. The model was fitted using MAP estimation (cf. Parameter estimation) and analysed how well the generative parameters (generating parameters in *Figure 5*) correlated with the recovered ones (fitted parameters in *Figure 5*) using Pearson correlation (summarised in *Figure 5c*). In addition to the correlation we examined the spread (*Figure 4—figure supplement 3*) of the recovered parameters. Overall the parameters were well recoverable.

## Model validation

To validate our model, we used each subjects' fitted parameters to simulate behaviour on our task (4000 trials per agent). The stimulated data (*Figure 5—figure supplement 3*), although not perfect, resembles the real data reasonably well. Additionally, to validate the behavioural indicators of the two different exploration heuristics we stimulated the behaviour of 200 agents using the winning model on one horizon condition (i.e. trials = 200). For the indicators of value-free random exploration, we stimulated behaviour with low ($\epsilon = 0$) and high ($\epsilon = 0.2$) values of the $\epsilon$-greedy parameter. The other parameters were set to the mean parameter fits ($\sigma_0 = 1.312$, $\eta = 2.625$, $Q_0 = 3.2$). This confirms that higher amounts of value-free random exploration are captured by the proportion of low-value bandit selection (*Figure 1f*) and the choice consistency (*Figure 1e*). Similarly, for the indicator of novelty exploration, we simulated behaviour with low ($\eta = 0$) and high ($\eta = 2$) values of the novelty bonus $\eta$ to validate the use of the proportion of the novel-bandit selection (*Figure 1g*). Again, the remaining parameters were set to the mean parameter fits ($\sigma_0 = 1.312$, $\epsilon = 0.1$, $Q_0 = 3.2$). Parameter values for high and low exploration were selected empirically from pilot and task data. Additionally, we simulated the effects of other exploration strategies in short and long horizon conditions (*Figure 1—figure supplement 3–5*). To simulate a long (versus short) horizon condition,we increased the overall exploration by increasing other exploration strategies. Details about parameter values can be found in *Appendix 2—table 7*.

## Acknowledgements

MD is a predoctoral fellow of the International Max Planck Research School on Computational Methods in Psychiatry and Ageing Research. The participating institutions are the Max Planck Institute for Human Development and the University College London (UCL). TUH is supported by a Wellcome Sir Henry Dale Fellowship (211155/Z/18/Z), a grant from the Jacobs Foundation (2017-1261-04), the Medical Research Foundation, a 2018 NARSAD Young Investigator Grant (27023) from the Brain and Behavior Research Foundation, and an ERC Starting Grant (946055). RJD holds a Wellcome Trust Investigator Award (098362/Z/12/Z). The Max Planck UCL Centre is a joint initiative supported by UCL and the Max Planck Society. The Wellcome Centre for Human Neuroimaging is supported by core funding from the Wellcome Trust (203147/Z/16/Z).

## Additional information

### Funding

| Funder | Grant reference number | Author |
| --- | --- | --- |
| Max-Planck-Gesellschaft | | Magda Dubois |
| Wellcome Trust | Sir Henry Dale Fellowship 211155/Z/18/Z | Tobias U Hauser |
| Jacobs Foundation | 2017-1261-04 | Tobias U Hauser |
| Wellcome Trust | Investigator Award 098362/Z/12/Z | Ray J Dolan |
| Medical Research Foundation | | Tobias U Hauser |
| Brain and Behavior Research Foundation | 27023 | Tobias U Hauser |
| European Research Council | 946055 | Tobias U Hauser |
| Wellcome Trust | Centre Award 203147/Z/16/Z | Ray J Dolan |

The funders had no role in study design, data collection and interpretation, or the decision to submit the work for publication.

### Author contributions

Magda Dubois, Conceptualization, Data curation, Software, Formal analysis, Writing - original draft, Writing - review and editing; Johanna Habicht, Jochen Michely, Data curation, Writing - review and editing; Rani Moran, Formal analysis, Writing - review and editing; Ray J Dolan, Funding acquisition, Writing - review and editing; Tobias U Hauser, Conceptualization, Software, Formal analysis, Supervision, Writing - original draft, Writing - review and editing

### Author ORCIDs

Magda Dubois https://orcid.org/0000-0002-5396-1855
Rani Moran https://orcid.org/0000-0002-7641-2402
Ray J Dolan https://orcid.org/0000-0001-9356-761X
Tobias U Hauser https://orcid.org/0000-0002-7997-8137

### Ethics

Human subjects: The study was approved by the UCL research committee (REC No 6218/002) and all subjects provided written informed consent.

### Decision letter and Author response

Decision letter https://doi.org/10.7554/eLife.59907.sa1
Author response https://doi.org/10.7554/eLife.59907.sa2

## Additional files

### Supplementary files
• Transparent reporting form

### Data availability
All necessary resources are publicly available at: https://github.com/MagDub/MFNADA-figures.

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

## Appendix 1

### Drug effect on response times

There were no differences in response times (RT) between drug groups in the one-way ANOVA. Neither in the mean RT (ANOVA: $F_{(2, 54)} = 1.625$, $p = 0.206$, $\eta^2 = 0.057$) nor in its variability (standard deviation; $F_{(2, 54)}=1.85$, $p = 0.16$, $\eta^2=0.064$).

### Bandit effect on response times

There was no difference in response times between bandits in the repeated-measures ANOVA (bandit main effect: $F_{(1.78, 99.44)} = 1.634$, $p = 0.203$, $\eta^2 = 0.028$; *Figure 3—figure supplement 1*).

### Interaction effects on response times

When looking at the first choice in both conditions, no differences were evident in RT in the repeated-measures ANOVA with a between-subject factor drug group and within-subject factors horizon and bandit (bandit main effect: $F_{(1.71, 92.46)} = 1.203$, $p = 0.3$, $\eta^2 = 0.022$; horizon main effect: $F_{(1, 54)} = 0.71$, $p = 0.403$, $\eta^2 = 0.013$; drug main effect: $F_{(2, 54)} = 2.299$, $p = 0.11$, $\eta^2 = 0.078$; drug-by-bandit interaction: $F_{(3.42, 92.46)} = 0.431$, $p = 0.757$, $\eta^2 = 0.016$; drug-by-horizon interaction: $F_{(2, 54)} = 0.204$, $p = 0.816$, $\eta^2 = 0.008$; bandit-by-horizon interaction: $F_{(1.39, 75.01)} = 0.298$, $p = 0.662$, $\eta^2 = 0.005$; drug-by-bandit-by-horizon interaction: $F_{(2.78, 75.01)} = 1.015$, $p = 0.387$, $\eta^2 = 0.036$).

In the long horizon, when looking at all six samples, no differences were evident in RT between drug group in the repeated-measures ANOVA with a between-subject factor drug group and within-subject factors bandits and samples (drug main effect: $F_{(2, 56)}=0.542$, $p = 0.585$, $\eta^2 = 0.019$). There was an effect of bandit (bandit main effect: $F_{(1.61, 90.12)}=7.137$, $p = 0.003$, $\eta^2 = 0.113$), of sample (sample main effect: $F_{(1.54, 86.15)} = 427.047$, $p<0.001$, $\eta^2 = 0.884$) and an interaction between the two (bandit-by-sample interaction: $F_{(3.33, 186.41)} = 4.789$, $p = 0.002$, $\eta^2 = 0.079$; drug-by-bandit interaction: $F_{(3.22, 90.12)} = 0.525$, $p = 0.679$, $\eta^2 = 0.018$; drug-by-sample interaction: $F_{(3.08, 86.15)} = 1.039$, $p = 0.381$, $\eta^2 = 0.036$; drug-by-bandit-by-sample interaction: $F_{(6.66, 186.41)} = 0.645$, $p = 0.71$, $\eta^2 = 0.023$). Further analysis (not corrected for multiple comparisons) revealed that the interaction between bandit and sample reflected the fact that when looking at samples individually, there was a bandit main effect in the second sample (bandit main effect: $F_{(1.27, 70.88)} = 27.783$, $p<0.001$, $\eta^2 = 0.332$; drug main effect: $F_{(2, 56)} = 0.201$, $p = 0.819$, $\eta^2 = 0.007$; drug-by-bandit interaction: $F_{(2.53, 70.88)} = 0.906$, $p = 0.429$, $\eta^2 = 0.031$) and in the third sample (bandit main effect: $F_{(1.23, 68.93)} = 21.318$, $p<0.001$, $\eta^2 = 0.276$; drug main effect: $F_{(2, 56)} = 0.102$, $p = 0.903$, $\eta^2 = 0.004$; drug-by-bandit interaction: $F_{(2.46, 68.93)} = 0.208$, $p = 0.855$, $\eta^2 = 0.007$), but not in the other samples first sample: drug main effect: $F_{(2, 56)} = 1.108$, $p = 0.337$, $\eta^2 = 0.038$; bandit main effect: $F_{(2, 112)} = 0.339$, $p = 0.713$, $\eta^2 = 0.006$; drug-by-bandit interaction: $F_{(4, 112)} = 0.414$, $p = 0.798$, $\eta^2 = 0.015$; fourth sample: (drug main effect: $F_{(2, 56)} = 0.43$, $p = 0.652$, $\eta^2 = 0.015$; bandit main effect: $F_{(1.36, 76.22)}=1.348$, $p = 0.259$, $\eta^2=0.024$; drug-by-bandit interaction: $F_{(2.72, 76.22)} = 0.396$, $p = 0.737$, $\eta^2 = 0.014$; fifth sample: drug main effect: $F_{(2, 56)} = 0.216$, $p = 0.806$, $\eta^2 = 0.008$; bandit main effect: $F_{(1.25, 69.79)}=0.218$, $p = 0.696$, $\eta^2 = 0.004$; drug-by-bandit interaction: $F_{(2.49, 69.79)} = 0.807$, $p = 0.474$, $\eta^2 = 0.028$; sixth sample: drug main effect: $F_{(2, 56)} = 1.026$, $p = 0.365$, $\eta^2 = 0.035$; bandit main effect: $F_{(1.05, 58.81)}=0.614$, $p = 0.444$, $\eta^2 = 0.011$; drug-by-bandit interaction: $F_{(2.1, 58.81)} = 1.216$, $p = 0.305$, $\eta^2 = 0.042$). In the second sample, the high-value bandit was chosen faster (high-value bandit vs low-value bandit: $t(59) = -5.736$, $p<0.001$, $d = 0.917$; high-value bandit vs novel bandit: $t(59) = -6.24$, $p<0.001$, $d = 0.599$) and the low-value bandit was chosen slower (low-value bandit vs novel bandit: $t(59) = 3.756$, $p<0.001$, $d = 0.432$). In the third sample, the low-value bandit was chosen slower (high-value bandit vs low-value bandit: $t(59) = -5.194$, $p<0.001$, $d = 0.571$; low-value bandit vs novel bandit: $t(59) = 4.448$, $p<0.001$, $d = 0.49$; high-value bandit vs novel bandit: $t(59) = -1.834$, $p = 0.072$, $d = 0.09$).

## Horizon effect on response times

There were no differences in RT between horizon conditions in the repeated-measures ANOVA with the between-subject factor drug group, the within-subject factor horizon condition and the covariates WASI and PANAS negative score horizon main effect: $F(1, 54) = 1.443$, $p = 0.235$, $\eta^2 = 0.026$; drug main effect: $F(2, 54) = 1.625$, $p = 0206$, $\eta^2 = 0.057$; drug-by-horizon interaction: $F(2, 54) = 0.431$, $p = 0.652$, $\eta^2 = 0.016$. In the long horizon, the RT decreased with each sample (sample main effect: $F(1.36, 73.5) = 13.626$, $p<0.001$, $\eta^2 = 0.201$; Pairwise comparisons: sample 1 vs 2: $t(59) = 20.968$, $p<0.001$, $d = 2.73$; sample 2 vs 3: $t(59) = 11.825$, $p<0.001$, $d = 1.539$; sample 3 vs 4: $t(59) = 7.862$, $p<0.001$, $d = 1.024$; sample 4 vs 5: $t(59) = 4.117$, $p<0.001$, $d = 1.539$; sample 5 vs 6: $t(59) = 2.646$, $p = 0.01$, $d = 1.024$; *Figure 2—figure supplement 1b*).

## PANAS

The Positive Affect and Negative Affect scale (PANAS; *Watson et al., 1988a*) was completed 50 min after the second drug administration and 10 min prior to the task. Groups had similar positive affect but differed in negative affect (*Appendix 2—table 1*), driven by a higher score in the placebo group (pairwise comparisons: placebo vs propranolol: $t(56)=2.801$, $p = 0.007$, $d = 0.799$; amisulpride vs placebo: $t(56)=-2.096$, $p = 0.041$, $d = 0.557$; amisulpride vs propranolol: $t(56) = 0.669$, $p = 0.506$, $d = 0.383$). It is unclear whether this difference was driven by the drug manipulation, but similar studies have not reported such an effect (e.g. *Hauser et al., 2019*; *Hauser et al., 2018*; *Campbell-Meiklejohn et al., 2011*; *Rogers et al., 2004*; *Hauser et al., 2017b*). We controlled for a possible influence of these measures in all our analyses.

## Physiological effects

Heart rate, systolic and diastolic pressure were obtained at 3 time points: at the beginning of the experiment before giving the drug ('at arrival'), after giving the drug just before the task ('pre-task'), and after finishing task and questionnaires ('post-task'). The post-task heart rate was lower for participants who received propranolol compared to the other two groups (one-way ANOVA: $F(2, 55) = 7.249$, $p = 0.002$, $\eta^2 = 0.209$; *Appendix 2—table 2*). A two-way ANOVA with the between-subject factor of drug group and within-subject factor of time (all three time points), showed a time-dependent decrease in heart rate ($F(1.74, 95.97) = 99.341$, $p<0.001$, $\eta^2 = 0.644$), in systolic pressure ($F(2, 110) = 8.967$, $p<0.001$, $\eta^2 = 0.14$) and in diastolic pressure ($F(2, 110) = 0.874$, $p = 0.42$, $\eta^2 = 0.016$), indicating subjects relaxed across the course of the study. Those reductions did not differ between drug group (drug main effect: heart rate: $F(2, 55) = 1.84$, $p = 0.169$, $\eta^2 = 0.063$; systolic pressure: $F(2, 55)=1.08$, $p = 0.347$, $\eta^2 = 0.038$; diastolic pressure: $F(2, 55) = 0.239$, $p = 0.788$, $\eta^2 = 0.009$; drug-by-time interaction: heart rate: $F(3.49, 95.97) = 1.928$, $p = 0.121$, $\eta^2 = 0.066$; systolic pressure: $F(4, 110) = 1.6$, $p = 0.179$, $\eta^2 = 0.055$; diastolic pressure: $F(4, 110) = 0.951$, $p = 0.438$, $\eta^2 = 0.033$).

## Task performance score

The performance did not differ between drug groups (total score: drug main effect: $F(2, 5) = 2.313$, $p = 0.109$, $\eta^2 = 0.079$) but it was increased in subjects with higher IQ scores (WASI main effect: $F(1, 54) = 17.172$, $p<0.001$, $\eta^2 = 0.241$).

In the long horizon, the score increased with each sample (sample main effect: $F(3.12, 174.97) = 103.469$, $p<0.001$, $\eta^2 = 0.649$; Pairwise comparisons: sample 1 vs 2: $t(59) = -6.737$, $p<0.001$, $d=0.877$; sample 2 vs 3: $t(59)=-3.69$, $p<0.001$, $d=0.48$; sample 3 vs 4: $t(59) = -5.167$, $p<0.001$, $d = 0.673$; sample 4 vs 5: $t(59) = -2.832$, $p = 0.006$, $d = 0.48$; sample 5 vs 6: $t(59) = -2.344$, $p = 0.022$, $d = 0.673$; *Figure 2—figure supplement 1a*). The increase in reward was larger in trials where the first draw was exploratory (linear regression slope coefficient: mean=0.118, sd=0.038) compared to when it was exploitative (linear regression slope coefficient: mean = 0.028, sd = 0.041; t-tests for slope coefficients: $t(58) = -12.161$, $p<0.001$, $d = -1.583$; *Figure 2—figure supplement 1d*), suggesting that exploration was used beneficially and subjects benefitted from their initial exploration.

## Dopamine effect on high-value bandit sampling frequency

The amisulpride group had a marginal tendency towards selecting the high-value bandit, meaning that they were disposed to exploit more overall (propranolol group excluded: horizon main effect: F (1, 35) = 3.035, p = 0.09, $\eta^2$ = 0.08; drug main effect: F(1, 35) = 3.602, p = 0.066, $\eta^2$ = 0.093; drug-by-horizon interaction: F(1, 35)=2.15, p = 0.151, $\eta^2$ = 0.058). This trend effect was not observed when all three groups were included (horizon main effect: F(1, 54) = 3.909, p = 0.053, $\eta^2$ = 0.068; drug main effect: F(2, 54) = 1.388, p = 0.258, $\eta^2$ = 0.049; drug-by-horizon interaction: F(2, 54) = 0.834, p = 0.44, $\eta^2$ = 0.03).

## Gender effects

When adding gender as a between-subjects variable in the repeated-measures ANOVAs, none of the main results changed. Interestingly, we observed a drug-by-gender interaction in the prior variance $\sigma_0$ (drug-by-gender interaction: F(2, 51) = 5.914, p = 0.005, $\eta^2$ = 0.188; *Figure 5—figure supplement 2*), driven by the fact that, female subjects in the placebo group had a larger average $\sigma_0$ (across both horizon conditions) compared to males (t(20) = 2.836, p = 0.011, d = 1.268), whereas male subjects have a larger $\sigma_0$ compared to females in the amisulpride group, (t(19) = -2.466, p = 0.025, d = 1.124; propranolol group: t(20) = -0.04, p = 0.969, d = 0.018). This suggests that in a placebo setting, females are on average more uncertain about an option's expected value, whereas in a dopamine blockade setting males are more uncertain. Besides this effect, we observed a trend-level significance in response times (RT), driven primarily by female subjects tending to have a faster RT in the long horizon compared to male subjects (gender main effect: F(1, 51) = 3.54, p = 0.066, $\eta^2$ = 0.065).

## Horizon and drug effects without covariate

When analysing the results without correcting for IQ (WASI) and negative affect (PANAS), similar results are obtained. The high-value bandit is picked more in the short-horizon condition indicating exploitation (F(1, 56) = 44.844, p<0.001, $\eta^2$ = 0.445), whereas the opposite phenomenon is observed in the low-value bandit (F(1, 56) = 24.24, p<0.001, $\eta^2$ = 0.302) and the novel bandit (horizon main effect: F(1, 56) = 30.867, p<0.001, $\eta^2$ = 0.355), indicating exploration. In line with these results, the model parameters for value-free random exploration ($\epsilon$: F(1, 56) = 10.362, p = 0.002, $\eta^2$ = 0.156) and novelty exploration ($\eta$: F(1, 56) = 38.103, p<0.001, $\eta^2$ = 0.405) are larger in the long compared to the short horizon condition. Additionally, noradrenaline blockade reduces value-free random exploration as can be seen in the two behavioural signatures, frequency of picking the low-value bandit (F(2, 56) = 2.523, p = 0.089, $\eta^2$ = 0.083; Pairwise comparisons: placebo vs propranolol: t(40)=2.923, p = 0.005, d=0.654; amisulpride vs placebo: t(38) = -0.587, p = 0.559, d = 0.133; amisulpride vs propranolol: t(38) = 2.171, p = 0.034, d = 0.496), and in the consistency (F(2, 56) = 3.596, p = 0.034, $\eta^2$ = 0.114; Pairwise comparisons: placebo vs propranolol: t(40) = -3.525, p = 0.001, d = 0.788; amisulpride vs placebo: t(38) = 1.107, p = 0.272, d = 0.251; amisulpride vs propranolol: t(38) = -2.267, p = 0.026, d = 0.514), as well as in the model parameter for value-free random exploration ($\epsilon$: F(2, 56) = 3.205, p = 0.048, $\eta^2$ = 0.103; Pairwise comparisons: placebo vs propranolol: t(40) = 3.177, p = 0.002, d = 0.71; amisulpride vs placebo: t(38) = 0.251, p = 0.802, d = 0.057; amisulpride vs propranolol: t(38) = 2.723, p = 0.009, d = 0.626).

## Horizon and drug effects with heart rate as covariate

When analysing results but now correcting for the post-experiment heart rate (*Appendix 2—table 1*) in addition to IQ (WASI) and negative affect (PANAS), we obtained similar results. Noradrenaline blockade reduced value-free random exploration as seen in two behavioural signatures, frequency of picking the low-value bandit (F(2, 52) = 4.014, p = 0.024, $\eta^2$ = 0.134; Pairwise comparisons:placebo vs propranolol: t(40) = 2.923, p = 0.005, d=0.654; amisulpride vs propranolol: t(38) = 2.171, p = 0.034, d = 0.496; amisulpride vs placebo: t(38) = -0.587, p = 0.559, d = 0.133), and consistency (F(2, 52) = 5.474, p = 0.007, $\eta^2$ = 0.174; Pairwise comparisons: placebo vs propranolol: t(40) = -3.525, p = 0.001, d=0.788; amisulpride vs propranolol: t(38) = -2.267, p = 0.026, d = 0.514; amisulpride vs

placebo: t(38) = 1.107, p = 0.272, d = 0.251), as well as in a model parameter for value-free random exploration ($\epsilon$: F(2, 52) = 4.493, p = 0.016, $\eta^2$ = 0.147; Pairwise comparisons: placebo vs propranolol: t(40) = 3.177, p = 0.002, d = 0.71; amisulpride vs propranolol: t(38) = 2.723, p = 0.009, d = 0.626; amisulpride vs placebo: t(38) = 0.251, p = 0.802, d = 0.057).

## Other model results

When analysing the fitted parameter values of both the second winning model (UCB + $\epsilon$ + $\eta$) and third winning model (hybrid + $\epsilon$ + $\eta$), similar results pertain. Thus, a value-free random exploration parameter was reduced following noradrenaline blockade in the second winning model ($\epsilon$: F(2, 54) = 4.503, p = 0.016, $\eta^2$ = 0.143; Pairwise comparisons: placebo vs propranolol: t(38)=2.185, p = 0.033, d=0.386; amisulpride vs propranolol: t(40) = 1.724, p = 0.089, d = 0.501; amisulpride vs placebo: t(40) = -0.665, p = 0.508, d = 0.151) and was affected at a trend-level significance in the third winning model ($\epsilon$: F(2, 54) = 3.04, p = 0.056, $\eta^2$ =0.101). These results highlight our finding that value-free random exploration is modulated by noradrenaline and additionally demonstrates this is independent of the complex exploration strategy used as well as the value function.

## Bandit combination effect

Behavioural results were analysed additionally for each bandit combination separately. The high-value bandit was chosen more when there was no novel bandit (pairwise comparisons: [certain-standard, standard, low] vs [certain-standard, standard, novel]: t(59) = 15.122, p<0.001, d = 1.969; [certain-standard, standard, low] vs [certain-standard, novel, low]: t(59) = 12.905, p<0.001, d = 2.389; [certain-standard, standard, low] vs [standard, novel, low]: t(59) = 18.348, p<0.001, d = 1.68), and less when its value was less certain ([standard, novel, low] vs [certain-standard, standard, novel]: t(59) = -6.986, p<0.001, d=0.407; [standard, novel, low] vs [certain-standard, novel, low]: t(59) = -5.44, p<0.001, d = 0.708; bandit combination main effect: F(1.81, 101.33) = 237.051, p<0.001, $\eta^2$ = 0.809; [certain-standard, standard, novel] vs [certain-standard, novel, low]: t(59) = 0.364, p = 0.717, d = 0.909; *Figure 3—figure supplement 2a*). The novel bandit was chosen most often when the high-value bandit was less certain, then when the high-value bandit was more certain and was chosen least when both certain and certain standard bandits were present ([standard, novel, low] vs [certain-standard, novel, low]: t(59)=5.001, p<0.001, d=0.651; [standard, novel, low] vs [certain-standard, standard, novel]: t(59) = 9.414, p<0.001, d = 1.226; [certain-standard, novel, low] vs [certain-standard, standard, novel]: t(59) = 4.146, p<0.001, d=0.54; bandit combination main effect: F(2, 112) = 42.44, p<0.001, $\eta^2$ = 0.431; *Figure 3—figure supplement 2b*). The low-value bandit was chosen less when the high-value bandit was more certain ([certain-standard, novel, low] vs [certain-standard, standard, low]: t(59) = -2.731, p = 0.008, d = 0.356; [certain-standard, novel, low] vs [standard, novel, low]: t(59) = -1.958, p = 0.055, d = 0.255; bandit combination main effect: F(1.66, 92.74) = 4.534, p = 0.019, $\eta^2$ = 0.075; [certain-standard, standard, low] vs [standard, novel, low]: t(59) = 1.32, p = 0.192, d = 0.172; *Figure 3—figure supplement 2c*).

## Other effects on choice consistency

Our results demonstrate a drug-by-horizon interaction on choice consistency (F(2, 54) = 3.352, p = 0.042, $\eta^2$ = 0.110; *Figure 3*), mainly driven by the fact that frequency of selecting the same option is increased in the long (compared to the short) horizon in the amisulpride group, while there is no significant horizon difference in the other two drug groups (pairwise comparison for horizon effect: amisulpride group: t(19) = 2.482, p = 0.023, d = 0.569; propranolol group: t(20) = -1.91, p = 0.071, d = 0.427; placebo group: t(20) = 0.505, p = 0.619, d = 0.113). It is not entirely clear why catecholamines would increase the differentiation between the horizon conditions and this relatively weak effect should be replicated before interpreting.

## Stand-alone heuristic models

We also analysed stand-alone heuristic models, in which there is no value computation (value of each bandit i: $V_i = 0$). The held-out data likelihood for such heuristic model combined with novelty exploration had a mean of m = 0.367 (sd = 0.005). The model in which we added value-free random

exploration on top of novelty exploration had a mean of m=0.384 (sd = 0.006). These models performed poorly, although better than chance level. Importantly, adding value-free random exploration improved performance. This highlights that subjects' combine complex and heuristic modules in exploration.

## Appendix 2

**Appendix 2—table 1.** Characteristics of drug groups.
The drug groups did not differ in gender, age, nor in intellectual abilities (adapted WASI matrix test).

Groups differed in negative affect (PANAS), driven by a higher score in the placebo group (pairwise comparisons: placebo vs propranolol: $t(56) = 2.801$, $p = 0.007$, $d = 0.799$; amisulpride vs placebo: $t(56) = -2.096$, $p = 0.041$, $d = 0.557$; amisulpride vs propranolol: $t(56) = 0.669$, $p = 0.506$, $d = 0.383$). For more details cf. Appendix 1. Mean (SD).

| | Propranolol | Placebo | Amisulpride | |
|---|---|---|---|---|
| Gender (M/F) | 10/10 | 10/10 | 10/9 | |
| Age | 22.80 (3.59) | 23.80 (4.23) | 23.05 (3.01) | $F(2,56) = 0.404$, $p = 0.669$, $\eta^2 = 0.014$ |
| Intellectual abilities | 22.8 (1.85) | 22.6 (3.70) | 24.37 (2.45) | $F(2,56) = 2.337$, $p = 0.106$, $\eta^2 = 0.077$ |
| Positive affect | 24.55 (8.99) | 28.90 (7.56) | 29.58 (10.21) | $F(2,56) = 1.832$, $p = 0.170$, $\eta^2 = 0.061$ |
| Negative affect | 10.65 (.81) | 12.75 (3.63) | 11.16 (1.71) | $F(2,56) = 4.259$, $p = 0.019$, $\eta^2 = 0.132$ |

**Appendix 2—table 2.** Physiological effects on drug groups.
The drug groups also differed in post-experiment heart rate, driven by lower values in the propranolol group (pairwise comparisons: placebo vs propranolol: $t(55)=3.5$, $p = 0.001$, $d = 1.293$; amisulpride vs placebo: $t(55) = -0.394$, $p = 0.695$, $d = 0.119$; amisulpride vs propranolol: $t(55)=3.013$, $p = 0.004$, $d = 0.921$). For detailed statistics and analysis accounting for this cf. Appendix 1. Mean (SD).

| | | Propranolol | Placebo | Amisulpride | |
|---|---|---|---|---|---|
| Heart rate (BPM) | At arrival | 74.9 (10.8) | 77,2 (12,6) | 77.7 (13.8) | $F(2, 55) = 0.290$, $p = 0.749$, $\eta^2 = 0.010$ |
| | Pre-task | 62,6 (8,5) | 65,8 (8,3) | 64,6 (9,8) | $F(2, 55) = 0.667$, $p = 0.517$, $\eta^2 = 0.024$ |
| | Post-task | 55,7 (6,7) | 64,4 (6,9) | 63,4 (10,0) | $F(2, 55) = 7.249$, $p = 0.002$, $\eta^2 = 0.209$ |
| Systolic blood pressure | At arrival | 117,2 (10,4) | 115,0 (9,7) | 117,9 (9,7) | $F(2, 55) = 0.438$, $p = 0.648$, $\eta^2 = 0.016$ |
| | Pre-task | 109,4 (9,2) | 111,8 (8,6) | 114,9 (8,6) | $F(2, 55) = 1.841$, $p = 0.168$, $\eta^2 = 0.063$ |
| | Post-task | 109,5 (8,2) | 113,9 (11,3) | 114,6 (9,3) | $F(2, 55) = 1.584$, $p = 0.214$, $\eta^2 = 0.054$ |
| Diastolic blood pressure | At arrival | 71,5 (7,8) | 71,2 (6,7) | 72,3 (6,7) | $F(2, 55) = 0.115$, $p = 0.891$, $\eta^2 = 0.004$ |
| | Pre-task | 68,3 (7,0) | 71,1 (10,6) | 72,0 (5,9) | $F(2, 55) = 1.111$, $p = 0.337$, $\eta^2 = 0.039$ |
| | Post-task | 70,8 (7,3) | 70,9 (8,0) | 70,3 (6,6) | $F(2, 55) = 0.037$, $p = 0.964$, $\eta^2 = 0.001$ |

**Appendix 2—table 3.** Table of statistics and behavioural values of *Figure 2*.
All of those measures were modulated by the horizon condition.

| | Horizon | Mean (sd) | Two-way repeated-measures ANOVA |
| --- | --- | --- | --- |
| | | | Main effect of horizon |
| Expected value | Short | 6.368 (0.335) | F(1, 56) = 19.457, p<0.001, $\eta^2$ = 0.258 |
| | Long | 6.221 (0.379) | |
| Initial samples | Short | 1.282 (0.247) | F(1, 56) = 58.78, p<0.001, $\eta^2$ = 0.512 |
| | Long | 1.084 (0.329) | |
| Score (first sample) | Short | 5.904 (0.192) | F(1, 56) = 58.78, p<0.001, $\eta^2$ = 0.512 |
| | Long | 5.82 (0.182) | |
| Score (average) | Short | 5.904 (0.192) | F(1, 56) = 103.759, p<0.001, $\eta^2$ = 0.649 |
| | Long | 6.098 (0.222) | |

**Appendix 2—table 4.** Table of statistics and behavioural measure values of *Figure 3*.
The drug groups differed in low-value bandit picking frequency (pairwise comparisons: placebo vs propranolol: t(40) = 2.923, p = 0.005, d = 0.654; amisulpride vs placebo: t(38) = -0.587, p = 0.559, d = 0.133; amisulpride vs propranolol: t(38)=2.171, p = 0.034, d = 0.496) and choice consistency (placebo vs propranolol: t(40) = -3.525, p = 0.01, d = 0.788; amisulpride vs placebo: t(38) = 1.107, p = 0.272, d = 0.251; amisulpride vs propranolol: t(38) = -2.267, p = 0.026, d = 0.514). The main effect is either of drug group (D) or of horizon (H). The interaction is either drug-by-horizon (DH) or horizon-by-WASI (measure of IQ; HW).

| | | Mean (sd) | | | Two-way repeated-measures ANOVA | | | |
| --- | --- | --- | --- | --- | --- | --- | --- | --- |
| | Horizon | Amisulpride | Placebo | Propranolol | Main effect | | Interaction | |
| High-value bandit | Short | 54.55 (8.87) | 49.38 (9.10) | 50.98 (11.4) | D | F(2, 54) = 1.388, p = 0.258, $\eta^2$ = 0.049 | DH | F(2, 54) = 0.834, p = 0.440, $\eta^2$ = 0.030 |
| | Long | 41.90 (8.47) | 44.10 (13.88) | 41.90 (13.57) | H | F(1, 54) = 3.909, p = 0.053, $\eta^2$ = 0.068 | HW | F(1, 54) = 13.304, p = 0.001, $\eta^2$ = 0.198 |
| Low-value bandit | Short | 3.32 (2.33) | 4.28 (2.98) | 2.50 (2.48) | D | F(2, 54) = 7.003, p = 0.002, $\eta^2$ = 0.206 | DH | F(2, 54) = 2.154, p = 0.126, $\eta^2$ = 0.074 |
| | Long | 5.45 (3.76) | 5.35 (3.40) | 3.45 (2.18) | H | F(1, 54) = 4.069, p = 0.049, $\eta^2$ = 0.070 | HW | F(1, 54) = 1.199, p = 0.278, $\eta^2$ = 0.022 |
| Novel bandit | Short | 36.87 (9.49) | 39.02 (10.94) | 40.15 (12.43) | D | F(2, 54) = 1.498, p = 0.233, $\eta^2$ = 0.053 | DH | F(2, 54) = 0.542, p = 0.584, $\eta^2$ = 0.020 |
| | Long | 46.82 (12.1) | 43.62 (16.27) | 48.55 (16.59) | H | F(1, 54) = 5.593, p = 0.022, $\eta^2$ = 0.094 | HW | F(1, 54) = 13.897, p<0.001, $\eta^2$ = 0.205 |
| Consistency | Short | 64.16 (12.27) | 62.70 (12.59) | 73.00 (11.33) | D | F(2, 54) = 7.154, p = 0.002, $\eta^2$ = 0.209 | DH | F(2, 54) = 3.352, p = 0.042, $\eta^2$ = 0.110 |
| | Long | 68.11 (10.34) | 64.00 (8.93) | 70.55 (9.91) | H | F(1, 54) = 1.333, p = 0.253, $\eta^2$ = 0.024 | HW | F(1, 54) = 0.409, p = 0.525, $\eta^2$ = 0.008 |

**Appendix 2—table 5.** Table of parameters used for each model compared during model selection (*Figure 4*).

Each of the 12 columns indicate a model. The three 'main models' studied were the Thompson model, the UCB model and a hybrid of both. Variants were then created by adding the $\epsilon$-greedy parameter, the novelty bonus and a combination of both. All the parameters besides $Q_0$ and w were fitted to each horizon separately. Parameters: $Q_0$ = prior mean (initial estimate of a bandits mean); $\sigma_0$ = prior variance (uncertainty about $Q_0$); w = contribution of UCB vs Thompson; $\gamma$ = information bonus; $\beta$ = softmax inverse temperature; $\epsilon$ = $\epsilon$-greedy parameter (stochasticity); $\eta$ = novelty bonus. Model selection measures include the cross-validation held-out data likelihood averaged over subjects, mean (SD), as well as the subject count for which this model performed better over either 12 models or over the 3 best models.

| | | Thompson | | | | UCB | | | | Hybrid | | | |
|---|---|---|---|---|---|---|---|---|---|---|---|---|---|
| | **Model** | | $+\epsilon$ | $+\eta$ | $+\epsilon+\eta$ | | $+\epsilon$ | $+\eta$ | $+\epsilon+\eta$ | | $+\epsilon$ | $+\eta$ | $+\epsilon+\eta$ |
| Parameters | Horizon independent | $Q_0$ | $Q_0$ | $Q_0$ | $Q_0$ | $Q_0$ | $Q_0$ | $Q_0$ | $Q_0$ | $w, Q_0$ | $w, Q_0$ | $w, Q_0$ | $w, Q_0$ |
| | Horizon dependent | $\sigma_0$ | $\sigma_0, \epsilon$ | $\sigma_0, \eta$ | $\sigma_0, \epsilon, \eta$ | $\gamma, \beta$ | $\gamma, \beta, \epsilon$ | $\gamma, \beta, \eta$ | $\gamma, \beta, \epsilon, \eta$ | $\sigma_0, \gamma, \beta$ | $\sigma_0, \gamma, \beta, \epsilon$ | $\sigma_0, \gamma, \beta, \eta$ | $\sigma_0, \gamma, \beta, \epsilon, \eta$ |
| Model selection | Mean held-out data likelihood | 550.2 (8.1) | 552.7 (7.1) | 552,2 (8.7) | 555.3 (8.4) | 552.9 (8.0) | 552.9 (8.0) | 553.4 (8.1) | 555.1 (8.8) | 553.5 (8.1) | 553.8 (8.4) | 555.0 (8.4) | 555.1 (8.5) |
| | Subjects for which model fits best (out of 12) | 0 | 3 | 2 | 20 | 0 | 0 | 1 | 20 | 0 | 0 | 7 | 6 |
| | Subjects for which model fits best (out of 3 best) | - | - | - | 27 | - | - | - | 22 | - | - | - | 10 |

**Appendix 2—table 6.** Table of statistics and fitted model parameters of *Figure 5*.

The drug groups differed in $\epsilon$-greedy parameter value (pairwise comparisons: placebo vs propranolol: t(40) = 3.177, p = 0.002, d =0 .71; amisulpride vs placebo: t(38) = 0.251, p = 0.802, d = 0.057; amisulpride vs propranolol: t(38) = 2.723, p = 0.009, d = 0.626). The main effect is either of drug group (D) or of horizon (H). The interaction is either drug-by-horizon (DH) or horizon-by-WASI (measure of IQ; HW).

| | | Mean (sd) | | | Two-way repeated-measures ANOVA | | | |
|---|---|---|---|---|---|---|---|---|
| | **Horizon** | **Amisulpride** | **Placebo** | **Propranolol** | **Main effect** | | **Interaction** | |
| $\epsilon$-greedy parameter | Short | 0.10 (0.10) | 0.12 (0.08) | 0.07 (0.08) | D | F(2, 54) = 6.722, p = 0.002, $\eta^2$ = 0.199 | DH | F(2, 54) = 1.305, p = 0.280, $\eta^2$ = 0.046 |
| | Long | 0.17 (0.14) | 0.14 (0.10) | 0.08 (0.06) | H | F(1, 54) = 1.968, p = 0.166, $\eta^2$ = 0.035 | HW | F(1, 54) = 6.08, p = 0.017, $\eta^2$ = 0.101 |

*Continued on next page*

*Appendix 2—table 6 continued*

| | Horizon | Mean (sd) | | | Two-way repeated-measures ANOVA | | | |
| | | Amisulpride | Placebo | Propranolol | Main effect | | Interaction | |
|---|---|---|---|---|---|---|---|---|
| Novelty bonus $\eta$ | Short | 2.07 (0.98) | 2.26 (1.37) | 2.05 (1.16) | D | $F(2, 54) = 0.249$, $p = 0.780$, $\eta^2 = 0.009$ | DH | $F(2, 54) = 0.03$, $p = 0.971$, $\eta^2 = 0.001$ |
| | Long | 3.24 (1.19) | 3.12 (1.63) | 2.95 (1.70) | H | $F(1, 54) = 1.839$, $p = 0.181$, $\eta^2 = 0.033$ | HW | $F(1, 54) = 8.416$, $p = 0.005$, $\eta^2 = 0.135$ |
| Prior variance $\sigma_0$ | Short | 1.18 (0.20) | 1.12 (0.43) | 1.25 (0.34) | D | $F(2, 54) = 0.060$, $p = 0.942$, $\eta^2 = 0.002$ | DH | $F(2, 54) = 2.162$, $p = 0.125$, $\eta^2 = 0.074$ |
| | Long | 1.41 (0.61) | 1.42 (0.59) | 1.21 (0.44) | H | $F(1, 54) = 0.129$, $p = 0.721$, $\eta^2 = 0.002$ | HW | $F(1, 54) = 0.022$, $p = 0.882$, $\eta^2 < 0.001$ |
| Prior mean $Q_0$ | | 3.22 (1.05) | 3.20 (1.36) | 3.44 (1.05) | D | $F(2, 54) = 0.118$, $p = 0.889$, $\eta^2 = 0.004$ | | |

**Appendix 2—table 7.** Parameter values used for simulations on *Figure 1—figure supplement 3–5*. Parameter values for high and low exploration were selected empirically from pilot and task data. Value-free random exploration and novelty exploration were simulated with an argmax decision function, which always selects the value with the highest expected value. For simulating the long (versus short) horizon condition, we assumed that not only the key value but also the other exploration strategies increased, as found in our experimental data. For each simulation $Q_0 = 5$ and unless otherwise stated, $\sigma_0 = 1.5$.

| | Horizon | Low exploration | High exploration | Additional parameters |
|---|---|---|---|---|
| Value-free random exploration | Short | $\epsilon = 0.1$ | $\epsilon = 0.2$ | $\eta = 0$ |
| | Long | $\epsilon = 0.3$ | $\epsilon = 0.4$ | $\eta = 2$ |
| Novelty exploration | Short | $\eta = 0$ | $\eta = 1$ | $\epsilon = 0$ |
| | Long | $\eta = 2$ | $\eta = 3$ | $\epsilon = 0.2$ |
| Thompson-sampling exploration | Short | $\sigma_0 = 0.8$ | $\sigma_0 = 1.2$ | $\eta = 0, \epsilon = 0$ |
| | Long | $\sigma_0 = 1.6$ | $\sigma_0 = 2$ | $\eta = 2, \epsilon = 0.2$ |
| UCB exploration | Short | $\gamma = 0.1$ | $\gamma = 0.3$ | $\beta = 5, \epsilon = 0$ |
| | Long | $\gamma = 0.7$ | $\gamma = 1.5$ | $\beta = 1.5, \epsilon = 0.2$ |

