## [Decision Letter]

**Acceptance summary:**

How individuals decide to exploit known options or to explore alternatives with unknown payouts is a fundamental question in neuroscience. This study combines human pharmacology and computational modeling to disentangle the role of two neurotransmitters (noradrenaline and dopamine) in driving different exploration strategies. Results show that value-independent "random" exploration heuristics are mediated by noradrenaline. These findings contribute to a better understanding of the neural processes involved in the exploration-exploitation trade-off.

**Decision letter after peer review:**

Thank you for submitting your article "Noradrenaline modulates tabula-rasa exploration" for consideration by *eLife*. Your article has been reviewed by Christian Büchel as the Senior Editor, a Reviewing Editor, and three reviewers. The following individual involved in review of your submission has agreed to reveal their identity: Christopher Warren (Reviewer #3).

The reviewers have discussed the reviews with one another and the Reviewing Editor has drafted this decision to help you prepare a revised submission.

Summary:

Dubois and colleagues investigate how two modes of exploration – tabula-rasa and novelty-seeking – contribute to human choice behavior. They found that subjects used both tabula-rasa and novelty-seeking heuristics when the task conditions were in favor of exploration. Specifically, participants could, and had to, make more responses in the long-horizon condition, which favored exploration, compared to the short-horizon condition, which favored exploitation. Moreover, the authors provide evidence that blockade of norepinephrine β receptors leads to decreased tabula-rasa exploration and increased choice consistency, whereas blockage of D2/D3 dopamine receptors had little effects.

All reviewers agreed that this paper provides interesting evidence on exploration-exploitation trade-offs and the underlying pharmacological mechanisms. However, reviewers felt that there are a number of major conceptual, methodological and interpretational issues that should be addressed in a revised version of the manuscript.

Essential revisions:

1) The term "tabula rasa" exploration is slightly misleading and using "random" exploration would be simpler, and clearer. That is, "tabula rasa" has the connotation that both the current "tabula rasa" choice and all future choices will not take into account information obtained before that choice. Random exploration is a better term because it is easy and intuitive to see that random choices can be sprinkled in with choices based on previous information, whereas "tabular rasa" implies wiping previous information away from that point forward. Indeed, previous related work has not termed the random exploration associated with the e-greedy parameter "tabula rasa". Problematic in this regard is that there is another parameter in one or more of the models that reflects random exploration (Subsection “Choice rules”, inverse temperature). This may be why the authors opted to call the e-greedy parameter something else. However, this raises the question: what is the difference between the e-greedy parameter and the inverse temperature mathematically, but more importantly, conceptually? At the very least, it would be important to provide a better explanation of the choice of term (tabula rasa) as well as a thorough explanation of the difference between tabula rasa and random exploration. We also recommend changing the term used, but we are amenable to accepting an argument for keeping it.

2) It is one thing to come up with computational terms and model-based quantities correlating with behavior but a different one to show their psychological meaning. Did the trials with tabula-rasa exploration or novelty exploration differ in terms of response times from the other types of responses? Did participants report that they indeed intended to explore in the tabula-rasa exploration trials? On a related note, how do the authors distinguish random (tabula-rasa) exploration from making a mistake? From how the task was designed, choosing the low value option appears to receive a more natural interpretation as a mistake rather than as exploration because this option was clearly dominated by the other options and remained so within and across trials.

3) Relatedly, successful performance in the task is based on the ability to discriminate between different reward types and to select the one with the higher value. From the experimental design description, one can see that in order to do so, the subjects needed to distinguish between different apple sizes. In this regard, a question arises: how large was the difference between two adjacent apple sizes? Was it large enough so that after a visual inspection, the participant could easily understand that the apple size = 7 was less rewarding than the apple size = 8? Finally, since the task requires visual inspection of reward stimuli, was the subject vision somehow tested and did it differ between groups?

4) Previous research of the authors (Hauser et al., 2017, 2018, 2019) has associated β receptor blockade with enhanced metacognition, decreased information gathering/increased commitment to an early decision (Hauser et al., 2018) and an arousal (i.e., reward)-induced boost of processing stimuli. In addition, Rogers et al., (2004) suggest that propranolol affects the processing of possible losses in decision-making paradigms, and might also reduce the discrimination between the different levels of possible gains (Rogers et al. 2004). In another study, Sokol-Hessner et al., (2015) also report a loss aversion reduction after propranolol administration. These effects might also change prior information and reset behavioral adaption to look for new opportunities. In this latter study the authors also report a lack of effect of propranolol onto choice consistency, contrary to what the present study reports. How do the current results relate to these previous findings? Of course, it is possible that norepinephrine plays multiple roles, but it appears not exactly parsimonious to imbue it with a different role for each task tested. Are there some commonalities across these effects that could be explained with some common function(s)?

5) Previous studies have shown that propranolol significantly decreased heart rate (e.g. Rogers et al., 2004). Did the authors measure heart rate and can they control for the possibility that peripheral effects of the drug explain the findings? And what was the reason for not collecting pupil diameter data, contrary to the previous research of the authors? Relatedly, in terms of norepinephrine influence and given the distributions of β receptors, could the authors be more explicit about the relation of their work to potential mechanisms (e.g. Goldman-Rakic et al., 1990 or Waterhouse et al., 1982)?

6) One strength of the paper is that the authors compared several computational models. The model selection is presented in Figure 4 and in Figure 4—figure supplement 1, the authors provide additional information regarding the winning model that accounted best for the largest number of subjects in comparison with two other models, namely the UCB model (with novelty and greedy parameters) or hybrid (with novelty and greedy parameters). It would be useful for the reader to get a better sense about the number of subjects which results favored any given model (i.e. a more exhaustive picture). One could use the same table as the one presented as in the Appendix—table 2 with respective number of subjects for which the model achieved the best performance. In fact, as shown in Figure 4, the winning model does not look very different (at least visually) from other models such as UCB (with novelty and greedy parameters) or hybrid (with novelty parameter or novelty and greedy parameters) models. As such, it would be important to know whether the conclusion about the e-greedy parameter would hold true if other model with similar performance were tested e.g. with UCB model (with novelty and greedy parameters) or hybrid (with novelty and greedy parameters)?

7) Related to this issue, the point of heuristics from a psychological perspective is that they dispense with the need to use full-blown algorithmic calculations. However, in the present models, the heuristics are only added on top of these calculations and the winning model includes Thompson exploration. Stand-alone heuristic models would do the term more justice and one wonders how well a model would fare that includes only tabula rasa exploration and novelty exploration.

8) The simulations provide a nice intuition for understanding choice proportions from different models/strategies (Figure 1E and 1F). However, it would be helpful to provide simulated results for long and short horizons separately. Do the models make different predictions for the two horizons? Additionally, it would be helpful to also show the results from other models (i.e. the proportion of low value bandit chosen by novelty agent). These could be added in the supplement.

[Editors” note: further revisions were suggested prior to acceptance, as described below.]

Thank you for resubmitting your article "Human complex exploration strategies are extended via noradrenaline-modulated heuristics" for consideration by *eLife*. Your revised article has been reviewed by Christian Büchel as the Senior Editor, a Reviewing Editor, and three reviewers. The following individual involved in review of your submission has agreed to reveal their identity: Christopher Warren (Reviewer #3).

The reviewers have discussed the reviews with one another and the Reviewing Editor has drafted this decision to help you prepare a revised submission.

Summary:

The authors have been very responsive to the initial reviews and reviewers feel that the paper is much improved. However, a few points remain. Please address the remaining issues raised by reviewer #1 and #2 and submit a revised version of the manuscript.

Reviewer #1:

Thank you for a largely responsive revision. The paper is much improved. A few points remain:

Related to previous point 1, the argument in subsection “Probing the contributions of heuristic exploration strategies” does not seem to be entirely correct. The authors claim that "A second prediction is that choice consistency, across repeated trials, is substantially affected by value-free random exploration." However, consistency can also be affected by the softmax parameter. If β is higher then choice consistency is also lower. Also, I am a little bit confused about the simulation results in Figure 1—figure supplement 2E,F. Do both models predict that the consistency of selecting the low value bandit is higher than the consistency of selecting the high value bandit? In line with the argument that higher β also lead to more stochastic choices, I also wonder if that can be the reason why UCB and UCB+𝜖 are not that much different in likelihood.

Regarding previous point 2: Were response time differences between value-free exploration and exploitation trials larger in the long horizon than the short horizon condition (i.e., while there was no main effect of bandit, was there an interaction with horizon or trial within horizon and was there a three-way interaction with drug)? Moreover, the response to the mistake issue is not entirely satisfactory. If participants paid (gradually) less attention in the long horizon, then it would also be expected that they make more mistakes in the long horizon condition only.

Regarding previous point 8, it is great that the authors followed our suggestion to simulate all models in both the short and long horizon. However, these figures (Figure 1—figure supplement 3 to Figure 1—figure supplement 5) seem to be somewhat confusing. The problem may lie in the parameters selected for simulation. According to Appendix 2—table 7, the multiple parameters were varied among different models. But I thought they should keep most consistent and only vary the interesting one. For example, shouldn't 𝜂 be kept same or even be zero in the value free random exploration model to show how choices are vary as function of 𝜖? I think the numbers are selected such that the predictions favor the value free random exploration model. If, as authors said, UCB + 𝜖 + 𝜂 is almost as good as Thompson-sampling + 𝜖 + 𝜂, I don’t see what the predictions can be such dramatically different. That is how I interpret the statement that " For simulating the long (versus short) horizon condition, we assumed that not only the key value but also the other exploration strategies increased, as found in our experimental data." Anyway, I feel the simulation data is somehow misleading and need more explanation.

Reviewer #2:

The authors addressed all my comments and made substantial revisions that have strengthened the overall manuscript. Specifically, the new information in Appendix—table 4 with each model's performance and additional analyses on of the other "(close to best)" models further strengthens the authors' claim. The authors also clarified the results on heart rate, RT and PANAS questionnaire, providing additional results and discussing appropriately potential caveats. Further additions in the Discussion address potential mechanisms of propranolol on decision making. The only comment I have relates to the sentence that follows (in the Discussion): “In particular, the results indicate that under propranolol behaviour is more deterministic and less influenced by “task-irrelevant” distractions. This aligns with theoretical ideas, as well as recent optogenetic evidence (32), that propose noradrenaline infuses noise in a temporally targeted way (31). It also accords with studies implicating noradrenaline in attention shifts (for a review cf. (76)). Other theories of noradrenaline/catecholamine function can link to determinism (64, 65), although the hypothesized direction of effect is different (i.e. noradrenaline increases determinism)." Here, it is unclear to me how the authors define determinism and how either increasing or decreasing noradrenaline can increase determinism?

Apart from that, the manuscript is well-written and provides an interesting account about the role of neuromodulatory systems on the processes at play during exploration.

---

## [Author Response]

Summary:Dubois and colleagues investigate how two modes of exploration – tabula-rasa and novelty-seeking – contribute to human choice behavior. They found that subjects used both tabula-rasa and novelty-seeking heuristics when the task conditions were in favor of exploration. Specifically, participants could, and had to, make more responses in the long-horizon condition, which favored exploration, compared to the short-horizon condition, which favored exploitation. Moreover, the authors provide evidence that blockade of norepinephrine β receptors leads to decreased tabula-rasa exploration and increased choice consistency, whereas blockage of D2/D3 dopamine receptors had little effects.All reviewers agreed that this paper provides interesting evidence on exploration-exploitation trade-offs and the underlying pharmacological mechanisms. However, reviewers felt that there are a number of major conceptual, methodological and interpretational issues that should be addressed in a revised version of the manuscript.

We thank the editors and reviewers for their positive evaluation of our manuscript and appreciate the helpful suggestions. We have now addressed all raised concerns and conducted substantial additional analyses in light of these comments.

In short, we have conducted analysis of physiological measures and have analysed our data with the relevant covariates. We have also analysed, in greater depth, the second and third best models in the computational modelling. We have also tested new models, run substantial additional model simulations, behavioural analyses, and analysed reaction times.

Regarding the text, we have clarified the manuscript by replacing the name “tabula-rasa” exploration with “value-free random” exploration and adapting the drug group names according to the reviewers’ suggestions. Moreover, we now provide substantial data simulations/illustrations to illustrate further the different exploration mechanisms. Lastly, we have expanded the discussion taking account of all the reviewers’ constructive suggestions. Importantly, all new analyses support our original Results and we believe this strengthens further the paper. We report key new results in the revised manuscript and trust that it meets the rigorous standards of your journal.

Essential revisions:1) The term "tabula rasa" exploration is slightly misleading and using "random" exploration would be simpler, and clearer. That is, "tabula rasa" has the connotation that both the current "tabula rasa" choice and all future choices will not take into account information obtained before that choice. Random exploration is a better term because it is easy and intuitive to see that random choices can be sprinkled in with choices based on previous information, whereas "tabular rasa" implies wiping previous information away from that point forward. Indeed, previous related work has not termed the random exploration associated with the e-greedy parameter "tabula rasa". Problematic in this regard is that there is another parameter in one or more of the models that reflects random exploration (subsection “Choice rules”, inverse temperature). This may be why the authors opted to call the e-greedy parameter something else. However, this raises the question: what is the difference between the e-greedy parameter and the inverse temperature mathematically, but more importantly, conceptually? At the very least, it would be important to provide a better explanation of the choice of term (tabula rasa) as well as a thorough explanation of the difference between tabula rasa and random exploration. We also recommend changing the term used, but we are amenable to accepting an argument for keeping it.

We thank the reviewer for raising this relevant point. We apologise for using a potentially misleading term. We chose the term “tabula-rasa” to distinguish it from other forms of stochasticity, such as the modulation of an inverse temperature. We agree that the form of exploration we wish to describe here is a pure form of randomness, one that ignores all available information. However, because “random” exploration has previously been used for inverse temperature related exploration, we refrained from using this term in our original manuscript and instead chose “tabula-rasa” to emphasize the fact that prior beliefs were not taken into account. We agree that tabula-rasa may have a connotation about future use of information, which the reviewer rightly highlights.

On the above basis we think it best to change the terminology. We now refer to “value-free random exploration” in the revised version of the manuscript, as distinct from “value-based random exploration” (as captured by Thompson sampling or softmax temperature). We believe these revised terms adequately reflect the putative computational mechanisms, whilst also highlighting a difference between them.

We apologise if the distinction between these two forms of exploration was not clear in the original manuscript. Mathematically, the value-based random exploration is captured by scaling the inverse temperature with the expected values in a softmax algorithm. This means that this form of exploration is still guided by the value of the choice options (hence “value-based”), and it requires an agent to keep track of each expected value that is compared. In contrast, the 𝜖-greedy algorithm ignores choice option values completely in 𝜖% of the time (hence “value-free”). Importantly, the difference between the inverse temperature and the 𝜖-greedy parameter is that the former requires more cognitive resources. We tabulate a summary of the strategies:

**Table resptable1:** 

Exploration strategy	Value-based random exploration	Value-free random exploration
**Algorithm**	Softmax	𝜖-greedy
**Equation**	P (i)=eBVi∑xeBVi	P(i)={(1−∈)ifi=bestaction∈otherwise
**Free parameters**	𝑉: bandit’s mean 𝛽: inverse temperature	𝜖: 𝜖-greedy parameter
**Values to compute**	𝑛 (number of bandits)	0

Although eventually both strategies increase noise in the decision process, their effect differs. Changing the softmax inverse temperature (cf. Figure 1—figure supplement 2A) affects the slope of the sigmoid, while changing the 𝜖-greedy parameter affects the compression of the sigmoid (cf. Figure 1—figure supplement 2B). Conceptually, in a softmax (value-based random) exploration mode (cf. Figure 1—figure supplement 2C), as each bandit's expected value is taken into account, an agent will still favour the second best bandit (i.e. medium-value bandit) over one with an even lower value (i.e. low-value bandit) when injecting noise. In contrast, in an 𝜖-greedy (value-free random) exploration mode (cf. Figure 1—figure supplement 2D), bandits are explored equally often irrespective of their expected value. This also has a consequence for choice consistency, in value-based random exploration the second best option is most probably explored (i.e. choice is still somehow consistent; cf. Figure 1—figure supplement 2E) versus equal probability of exploring any of the non-optimal options in an 𝜖-greedy (value-free random) exploration mode (i.e. low consistency, cf. Figure 1—figure supplement 2F). Please also see our response to comment 6., where we demonstrate that our effects remain unchanged even when allowing value-based and value-free random exploration to directly compete.

In addition to changing the name of these exploration strategies in the revised manuscript, we now also provide a more thorough explanation of the difference between the two random exploration strategies and we illustrate the difference in the newly added Figure 1—figure supplement 2.

Introduction: “An alternative strategy, sometimes termed “random” exploration, is to induce stochasticity after value computations in the decision process. […] Of relevance in this context is a view that exploration strategies depend on dissociable neural mechanisms (21). Influences from noradrenaline and dopamine are plausible candidates in this regard based on prior evidence (9, 22).”

2) It is one thing to come up with computational terms and model-based quantities correlating with behavior but a different one to show their psychological meaning. Did the trials with tabula-rasa exploration or novelty exploration differ in terms of response times from the other types of responses? Did participants report that they indeed intended to explore in the tabula-rasa exploration trials? On a related note, how do the authors distinguish random (tabula-rasa) exploration from making a mistake? From how the task was designed, choosing the low value option appears to receive a more natural interpretation as a mistake rather than as exploration because this option was clearly dominated by the other options and remained so within and across trials.

We thank the reviewers for raising this important point, on which we now elaborate in more detail in the manuscript.

The key dissociation between value-free random exploration and simply making a mistake is that the former is sensitive to our task (horizon) condition, which would not be expected for mistakes. This means that pure “mistakes” (i.e. independent of any cognitive process) should be equally distributed across all experimental conditions, whereas value-free exploration is deployed more strategically, increasing exploration over a long horizon.

In our data, we find that value-free exploration increases over the long horizon, i.e. when exploration is more useful (low-value bandit: horizon main effect: F(1, 54)=4.069, p=.049, 𝜂^2^=.070; 𝜖-greedy parameter: horizon-by-WASI interaction: F(1, 54)=6.08, p=.017, 𝜂^2^=.101). Importantly, we since reproduced this horizon effect multiple times across independent studies and cohorts. We found the same horizon effect in children and adolescents (low-value bandit: horizon main effect: F(1, 94)=8.837, p=.004, 𝜂^2^=.086; 𝜖-greedy parameter: horizon main effect: F(1, 94)=20.63, p<.001, 𝜂^2^=.180; Dubois et al., 2020, BioRxiv) as well as in healthy adults online (unpublished pilot data: low-value bandit: t(61)=-3.621, p=.001, d=.46, 95%CI=[-1.615, -.466]; 𝜖-greedy parameter: pilot data: t(61)=-3.286, p=.002, d=.417, 95%CI=[-.058, -.014]). These results demonstrate that this form of exploration is modulated by horizon, which would not be the case if they were simple mistakes.

Based on the reviewers’ suggestion, we also investigated response times, based on a hypothesis that simple mistakes would lead to faster responses. We did not observe any response times difference between low-value bandit trials and trials in which the high-value bandit (i.e. exploitation) or the novel bandit were chosen (bandit main effect: F(1.78 , 99.44)=1.634 , p=.203 , 𝜂^2^=.028; Figure 3—figure supplement 1). This further speaks against a hypothesis that these choices represent mere mistakes.

Lastly, it is important to note our finding that value-free random exploration is modulated by propranolol. To our knowledge, previous studies using the same drug (Sokol-Hessner et al., 2015; Campbell-Meiklejohn et al., 2011; Rogers et al., 2004; Hauser et al., 2019) did not report any impact on behavioural features that could be interpreted as “mistakes”. For example, our own previous study did not find an effect of propranolol on choice accuracy (Hauser et al., 2019). Instead, in prior studies propranolol impacted a directed cognitive process, similar to what we find here.

In line with previous papers on exploration (e.g. Warren et al., 2017; Wilson et al., 2014; Wu et al., 2018; Stojic et al., 2020), for several reasons we did not collect subjects’ reports about their intentions. Firstly, we were concerned this could have biased the task as subjects might feel compelled to focus on exploration, rather than performing the task and earning as much money as possible. This means that exploration might be perceived as a means to satisfy an experimenter’s intentions, rather than harvesting information for later use. Second, it is unclear whether all forms of exploration invoke a conscious representation (and hence are accessible to self-reports). It is possible that associated heuristics might be phylogenetically old, and not represented explicitly. Lastly, seeking subject reports would have either extended the task duration substantially, or reduced the number of trials, both restrictions that we wished to avoid.

In the revised manuscript, we discuss the psychological meaning of these exploration strategies in more detail with a focus on the dissociation between mistakes and value-free random exploration. We also added the new response latency analyses.

Discussion: “Value-free random exploration might reflect other influences, such as attentional lapses or impulsive motor responses. […] However, future studies could explore these exploration strategies in more detail including by reference to subjects’ own self-reports.”

Appendix I: “There was no difference in response times between bandits in the repeated-measures ANOVA (bandit main effect: F(1.78 , 99.44)=1.634 , p=.203 , η2=.028; Figure 3—figure supplement 1).”

3) Relatedly, successful performance in the task is based on the ability to discriminate between different reward types and to select the one with the higher value. From the experimental design description, one can see that in order to do so, the subjects needed to distinguish between different apple sizes. In this regard, a question arises: how large was the difference between two adjacent apple sizes? Was it large enough so that after a visual inspection, the participant could easily understand that the apple size = 7 was less rewarding than the apple size = 8? Finally, since the task requires visual inspection of reward stimuli, was the subject vision somehow tested and did it differ between groups?

We agree, this is a relevant point and one we investigated in detail when developing the task. In fact, we originally tested versions with different apple size ranges and based upon this opted for a smaller range of 9 different apple sizes, as our pilots showed that they were easily distinguishable. Moreover, the apples were presented (and remained) next to each other on the screen in the “crate”, so that apple sizes were directly comparable.

Even though we did not assess the vision of our subjects formally, we only recruited subjects who had (self-reported) normal or corrected-to-normal vision. This is the standard procedure for participant recruitment at the Wellcome Centre for Human Neuroimaging. To assess subjects’ understanding of apple sizes and to confirm normal vision, we conducted extensive training prior to the main experiment, in which they had to categorise different apple sizes. This training was successfully completed by all participants. We have now added this information and Figure 1—figure supplement 1.

Materials and methods: “Each distribution was truncated to [2, 10], meaning that rewards with values above or below this interval were excluded, resulting in a total of 9 possible rewards (i.e. 9 different apple sizes; cf. Figure 1—figure supplement 1 for a representation). […] In training, based on three displayed apples of similar size, they were tasked to guess between two options, namely which apple was most likely to come from the same tree and then received feedback about their choice.”

4) Previous research of the authors (Hauser et al., 2017, 2018, 2019) has associated β receptor blockade with enhanced metacognition, decreased information gathering/increased commitment to an early decision (Hauser et al., 2018) and an arousal (i.e., reward)-induced boost of processing stimuli. In addition, Rogers et al., (2004) suggest that propranolol affects the processing of possible losses in decision-making paradigms, and might also reduce the discrimination between the different levels of possible gains (Rogers et al., 2004). In another study, Sokol-Hessner et al., (2015) also report a loss aversion reduction after propranolol administration. These effects might also change prior information and reset behavioral adaption to look for new opportunities. In this latter study the authors also report a lack of effect of propranolol onto choice consistency, contrary to what the present study reports. How do the current results relate to these previous findings? Of course, it is possible that norepinephrine plays multiple roles, but it appears not exactly parsimonious to imbue it with a different role for each task tested. Are there some commonalities across these effects that could be explained with some common function(s)?

We thank the reviewers for raising this interesting point about the overarching function of noradrenaline. As the referee indicates we previously associated β receptor blockade with enhanced metacognition, decreased information gathering and decreased arousal-induced boosts of processing stimuli (please note that these studies were conducted in different subjects and are thus not directly relatable). Together with our current findings, the overall pattern of results might suggest that propranolol impacts how neural noise affects information processing in the brain, in line with prior theoretical work (Dayan and Yu, 2006). In particular, all of these results show that by administering propranolol, behaviour is more deterministic and less influenced by “task-irrelevant distractions”, an observation that accords also with reports implicating noradrenaline in attention shifting (for a review cf. Trofimova and Robbins, 2016). For example, an arousal-induced boost in incidental memory is abolished after propranolol (Hauser et al., 2019), a finding that aligns well with the suggestion of Dayan and Yu (2006) that noradrenaline can infuse noise into a system in a temporally targeted way. The latter idea also gains support from recent optogenetic based studies (Tervo et al., 2014) and relates also to other theories of noradrenaline/catecholamine function (Servan-Schreiber et al., 1990; Aston-Jones and Cohen, 2005), although an assumption here was that increases in noradrenaline would lead to an increase in determinism.

The studies pointed out by the reviewers (Rogers et al., 2004; Sokol-Hessner et al., 2015) make an interesting point about loss processing, including a demonstration that propranolol attenuates processing of punishment cues (Rogers et al., 2004) and reduces loss aversion (Sokol-Hessner et al., 2015), suggesting an effect of noradrenaline on prior information in a loss context. As the referee will appreciate our task was conducted in a reward-context, and it is entirely possible that an exploration task in a loss setting would reveal additional interesting results. Our interpretation of the existing data is that propranolol has a minimal, if any, effect on levels of reward. Firstly, the above-mentioned study by Rogers et al., (2004) only found a trend-level result. In addition, unpublished data from our group (Habicht et al., in prep) has not revealed any effect of propranolol on representations of gain and reward magnitudes.

It is interesting to speculate why Sokol-Hessner et al., (2015) did not find an effect on consistency. We do not believe that the latter results question our findings. It is important to note we refer to as consistency as the number of times subjects made the same exact choice on the exact same trial. We built this into the design of our task by duplicating each trial. In the study by Sokol-Hessner et al., (2015), the authors defined consistency as the softmax temperature parameter in a non-exploration related context. In line with their findings, we did not observe any drug effect on our prior variance 𝜎_"_ parameter, the one most closely related to the parameter in the Sokol-Hessner et al., (2015) study.

We now incorporate these points into the revised version of the paper. We have clarified how we measure consistency and how this is different from other studies. We add a paragraph discussing the above papers and speculate on an overarching explanatory framework and how this might relate to the previous theories about the role of noradrenaline.

Discussion: “Noradrenaline blockade by propranolol has been shown previously to enhance metacognition (75), decrease information gathering (59), and attenuate arousal-induced boosts in incidental memory (36). […] Future studies investigating exploration in loss contexts might provide important additional information on these questions.”

Materials and methods: “[Trials] were then duplicated to measure choice consistency, defined as the frequency of making the same choice on identical trials (in contrast to a previous propranolol study where consistency was defined in terms of a value-based exploration parameter (60)).”

5) Previous studies have shown that propranolol significantly decreased heart rate (e.g. Rogers et al., 2004). Did the authors measure heart rate and can they control for the possibility that peripheral effects of the drug explain the findings? And what was the reason for not collecting pupil diameter data, contrary to the previous research of the authors? Relatedly, in terms of norepinephrine influence and given the distributions of β receptors, could the authors be more explicit about the relation of their work to potential mechanisms (e.g. Goldman-Rakic et al., 1990 or Waterhouse et al., 1982)?

We thank the reviewers for raising these points and suggesting these additional analyses. We recorded heart rate and blood pressure (systolic and diastolic) as part of our standard protocol to ensure subjects’ health and safety. Those were collected at 3 time points: at the beginning of the experiment before giving the drug (“at arrival”), after giving the drug just before playing the task (“pre-task”), and after finishing the experiment (“post-task”). We have now, as suggested, analysed these data.

In line with the known physiological effects of propranolol (Koudas et al., 2019) and previous cognitive studies (e.g. Rogers et al., 2004, Hauser et al., 2019), the propranolol group had a lower post-task heart rate (F(2, 55)=7.249, p=.002, 𝜂^2^=.209). None of the other measures and timepoints showed any drug effect (cf. Appendix 2 Table 2 for all comparisons).

To further evaluate these effects, we ran a two-way ANOVA with the between-subject factor drug group and the within-subject factor time (all three time points). In this analysis, we found a change in all measures for all subjects over time (heart rate: F(1.74, 95.97)=99.341, p<.001, 𝜂^2^=.644); systolic pressure: F(2, 110)=8.967, p<.001, 𝜂^2^=.14; diastolic pressure: F(2, 110)=.874, p=.42, 𝜂^2^=.016, meaning these measures decreased throughout the experiment. However, did not differ between groups (drug main effect: heart rate: F(2, 55)=1.84, p=.169, 𝜂^2^=.063; systolic pressure: F(2, 55)=1.08, p=.347, 𝜂^2^=.038; diastolic pressure: F(2, 55)=.239, p=.788, 𝜂^2^=.009; drug-by-time interaction: heart rate: F(3.49, 95.97)=1.928, p=.121, 𝜂^2^=.066; systolic pressure: F(4, 110)=1.6, p=.179, 𝜂^2^=.055; diastolic pressure: F(4, 110)=.951, p=.438, 𝜂^2^=.033).

To ensure a lower heart rate in the post experiment measurement did not impact our results, we reanalysed all our data by adding the post-experiment heart-rate as an additional covariate. We found that controlling for this peripheral marker did not alter any of our findings. In particular, we replicated the same drug effects for value-free random exploration, in the behavioural measures: frequency of picking the low-value bandit (main effect of drug: F(2, 52) = 4.014, p=.024, 𝜂^2^=.134; Pairwise comparisons: placebo vs propranolol: t(40) = 2.923, p=.005, d=.654; amisulpride vs propranolol: t(38) = 2.171, p=.034, d=.496; amisulpride vs placebo: t(38) = -.587, p=.559, d=.133) and choice consistency (F(2, 52) = 5.474, p=.007, 𝜂^2^=.174; Pairwise comparisons: placebo vs propranolol: t(40) = -3.525, p=.001, d=.788; amisulpride vs propranolol: t(38) = -2.267, p=.026, d=.514; amisulpride vs placebo: t(38) = 1.107, p=.272, d=.251), and also the modeling parameter 𝜖-greedy (F(2, 52) = 4.493, p=.016, 𝜂^2^=.147; Pairwise comparisons: placebo vs propranolol: t(40) = 3.177, p=.002, d=.71; amisulpride vs propranolol: t(38) = 2.723, p=.009, d=.626; amisulpride vs placebo: t(38)=.251, p=.802, d=.057). Thus, we believe our findings are unlikely to have arisen from peripheral effects of the drug. We now mention this in the discussion and report traditional analyses in the revised manuscript (cf. Appendix 1, Appendix 2—table 2).

We thank the reviewers for raising the question about pupillometry, an important topic which we believe is more complex than commonly perceived. We appreciate there is a lot of enthusiasm about pupil size as an indirect measure of noradrenaline function, and indeed animal recordings show a nice alignment between locus coeruleus firing and pupil size (e.g. Joshi et al., 2016). However, there are several issues with respect to human studies that remain unclear, including specificity, directionality and causality of these effects. We discuss this in detail in Hauser et al., (2019), and a summary of these limitations can be found in the recent review by Joshi and Gold, (2020). In short, it is important to highlight that the link between noradrenaline and pupil dilation remains unresolved (Nieuwenhuis et al., 2011). In fact, it has been suggested that noradrenaline does not directly drive pupil dilation, but instead both are driven by a common input (e.g. Nieuwenhuis et al., 2011). Therefore, even though pupillometry might reflect endogenous fluctuations in noradrenaline, pharmacologically induced changes of noradrenaline levels may have very different effects that remain poorly understood. In fact, our own, and others, previous studies have found no effect of propranolol on pupil diameter (Koudas et al., 2009; Hauser et al., 2019), but instead a significant effect of drugs other than noradrenaline (e.g. dopamine; Samuels et al., 2006; Hauser et al., 2019).

An additional reason for not including pupillometry was that pupillometry has strong restrictions in terms of task presentations (luminance matching, absence of eye gaze). In fact, our pilot studies showed that this current task was not feasible when applying these restrictions.

Based on these concerns, we decided against using pupillometry. We now discuss this in detail in the revised version of the manuscript, including highlighting limitations and elaborating on future applications.

Regarding β-receptors and potential mechanisms, we thank the reviewers for pointing out those interesting papers. Waterhouse et al., (1982) have shown that β-receptors increase synaptic inhibition specifically through inhibitory GABA-mediated transmission (Waterhouse et al., 1984). This in line with findings from Goldman-Rakic et al., (1990) who found that intermediate layers in the prefrontal areas, within which inhibition is favored (Isaacson et al., 2011), host a high concentration of β-receptors. All of these results suggest that noradrenaline-related task-distractibility, and randomness, could reflect inhibitory mechanisms. We discuss this in detail in the revised Discussion. Please also see our response 5., where we speculate about the specific receptor effects and the implications on our findings.

Materials and methods: “The groups consisted of 20 subjects each matched (cf. Appendix 2—table 1) for gender and age. To evaluate peripheral drug effects, heart rate, systolic and diastolic blood pressure were collected at three different time-points: “at arrival”, “pre-task” and “post-task”, cf. Appendix 1 for details.”

Results: “Similar results were obtained in an analysis that corrected for physiological effects as from the analysis without covariates (cf. Appendix 1).”

Discussion: “Because the effect of pharmacologically induced changes of noradrenaline levels on pupil size remains poorly understood (36, 67), including the fact that previous studies found no effect of propranolol on pupil diameter (36, 68), we opted against using pupillometry in this study. […] However, future studies using drugs that exclusively targets peripheral, but not central, noradrenaline receptors (e.g. (82)) are needed to answer this question conclusively.”

Appendix 1: “Heart rate, systolic and diastolic pressure were obtained at 3 time points: at the beginning of the experiment before giving the drug (“at arrival”), after giving the drug just before the task (“pre-task”), and after finishing task and questionnaires (“post-task”). […] When analysing results but now correcting for the post-experiment heart rate (cf. Appendix 2 Table 1) in addition to IQ (WASI) and negative affect (PANAS), we obtained similar results. Noradrenaline blockade reduced value-free random exploration as seen in two behavioural signatures, frequency of picking the low-value bandit (F(2, 52) = 4.014, p=.024, 𝜂^2^=.134; Pairwise comparisons:(placebo vs propranolol: t(40) = 2.923, p=.005, d=.654; amisulpride vs propranolol: t(38) = 2.171, p=.034, d=.496; amisulpride vs placebo: t(38) = -.587, p=.559, d=.133), and consistency F(2, 52) = 5.474, p=.007, 𝜂^2^=.174; Pairwise comparisons: placebo vs propranolol: t(40) = -3.525, p=.001, d=.788; amisulpride vs propranolol: t(38) = -2.267, p=.026, d=.514; amisulpride vs placebo: t(38) = 1.107, p=.272, d=.251), as well as in a model parameter for value-free random exploration (ϵ: F(2, 52) = 4.493, p=.016, 𝜂^2^=.147; Pairwise comparisons: placebo vs propranolol: t(40) = 3.177, p=.002, d=.71; amisulpride vs propranolol: t(38) = 2.723, p=.009, d=.626; amisulpride vs placebo: t(38)=.251, p=.802, d=.057).”

6) One strength of the paper is that the authors compared several computational models. The model selection is presented in Figure 4 and in Figure 4—figure supplement 1, the authors provide additional information regarding the winning model that accounted best for the largest number of subjects in comparison with two other models, namely the UCB model (with novelty and greedy parameters) or hybrid (with novelty and greedy parameters). It would be useful for the reader to get a better sense about the number of subjects which results favored any given model (i.e. a more exhaustive picture). One could use the same table as the one presented as in the Appendix—table 2 with respective number of subjects for which the model achieved the best performance. In fact, as shown in Figure 4, the winning model does not look very different (at least visually) from other models such as UCB (with novelty and greedy parameters) or hybrid (with novelty parameter or novelty and greedy parameters) models. As such, it would be important to know whether the conclusion about the e-greedy parameter would hold true if other model with similar performance were tested e.g. with UCB model (with novelty and greedy parameters) or hybrid (with novelty and greedy parameters)?

We thank the reviewer for acknowledging our efforts in terms of model selection. As suggested, we now expand on this section and provide additional details.

We now show that Thompson+𝜖+𝜂 model has the highest subject count when comparing the 3 best models. When comparing across all models, this same model is equally first with the UCB+𝜖+𝜂 model in subject count with a highest average likelihood of held-out data making it the winning model (N=20 for each model). We now show this in Figure 4—figure supplement 1 and we extended Appendix—table 4 with each model’s performance.

It is important to note that the purpose of the model comparison was to demonstrate the relevance of the two new heuristics, in addition to complex exploration strategies. The model comparison, as well as the new head counts, show clearly that the winning models all incorporate both the novelty and value-free exploration modules, strongly supporting our key message. The fact that the Thompson, UCB and hybrid model performed relatively similar may highlight that they make fairly similar predictions in our task, and that benefits of a particular complex model may be explained by these similar heuristic strategies. We discuss this now in more detail in the revised manuscript.

We thank the reviewers for suggesting these additional analyses of the other (close to best) models. We indeed find a very similar effect (i.e. reduction of value-free random exploration following propranolol) in the second winning model (UCB+𝜖+𝜂; drug main effect on 𝜖: F(2, 54)=4.503, p=.016, η2=.143) and (almost) in the third place model (hybrid+𝜖+𝜂; drug main effect on 𝜖: F(2, 54 )=3.04, p=.056, η2=.101). We believe that these effects further extend and support our findings and we now discuss them in the revised Discussion and provide detail in the Appendix.

Results: “Interestingly, although the second and third place models made different prediction about the complex exploration strategy, using a directed exploration with value-based random exploration (UCB) or a combination of complex strategies (hybrid) respectively, they share the characteristic of benefitting from value-free random and novelty exploration. This highlights that subjects used a mixture of computationally demanding and heuristic exploration strategies. […]. Critically, the effect on ϵ was also significant when the complex exploration strategy was a directed exploration with value-based random exploration (second place model) and, marginally significant, when it was a combination of the above (third place model; cf. Appendix 1).”

Discussion: “Importantly, these heuristics were observed in all best models (first, second and third position) even though each incorporated different exploration strategies. This suggests that the complex models made similar predictions in our task, and demonstrates that value-free random exploration is at play even when accounting for other value-based forms of random exploration (1, 7), whether fixed or uncertainty-driven. […] Importantly, this effect was observed whether the complex exploration was an uncertainty-driven value-based random exploration (winning model), a directed exploration with value-based random exploration (second place model) or a combination of the above (third place model; cf. Appendix 1).”

Appendix 1: “When analysing the fitted parameter values of both the second winning model (UCB + ϵ + η) and third winning model (hybrid + ϵ + η), similar results pertain. Thus, a value-free random exploration parameter was reduced following noradrenaline blockade in the second winning model (ϵ: F(2, 54)=4.503, p=.016, 𝜂^2^=.143; Pairwise comparisons: placebo vs propranolol: t(38)=2.185, p=.033, d=.386; amisulpride vs propranolol: t(40)=1.724, p=.089, d=.501; amisulpride vs placebo: t(40)=-.665, p=.508, d=.151) and was affected at a trend-level significance in the third winning model (ϵ: F(2, 54 )=3.04, p=.056, 𝜂^2^=.101).”

7) Related to this issue, the point of heuristics from a psychological perspective is that they dispense with the need to use full-blown algorithmic calculations. However, in the present models, the heuristics are only added on top of these calculations and the winning model includes Thompson exploration. Stand-alone heuristic models would do the term more justice and one wonders how well a model would fare that includes only tabula rasa exploration and novelty exploration.

We thank the reviewers for this suggestion. We based our hypotheses/analyses on recent evidence for complex exploration strategies and used model selection to show the presence of explorations heuristics in addition to these complex strategies. Based on the reviewers’ suggestions, we have added stand-alone heuristic models (value-free random exploration and novelty exploration with no value function computation; cf. Appendix 1).

As can be seen from the results in Appendix 1, these models performed poorly, although better than chance level, while adding value-free random exploration substantially improved their performance. Our results thus highlight that subjects combine complex and heuristic modules in exploration. We believe this is a valuable new insight and we have now added these additional models to the revised manuscript.

Appendix 1: “We also analysed stand-alone heuristic models, in which there is no value computation (value of each bandit i: 𝑉_i_ = 0). The held-out data likelihood for such heuristic model combined with novelty exploration had a mean of m=0.367 (sd=0.005). The model in which we added value-free random exploration on top of novelty exploration had a mean of m=0.384 (sd=0.006). These models performed poorly, although better than chance level. Importantly, adding value-free random exploration improved performance. This highlights that subjects’ combine complex and heuristic modules in exploration.”

8) The simulations provide a nice intuition for understanding choice proportions from different models/strategies (Figure 1E and 1F). However, it would be helpful to provide simulated results for long and short horizons separately. Do the models make different predictions for the two horizons? Additionally, it would be helpful to also show the results from other models (i.e. the proportion of low value bandit chosen by novelty agent). These could be added in the supplement.

We now extend the model simulations and agree that they can provide a more detailed understanding. To ensure an intuitive understanding for a general audience, we provide these additional simulations as 3 additional figures in the Figure supplement, and to keep the original simulation graphs as intuitive illustrations.

From the new figures, one can see that the frequency of picking the low-value bandit, as well as choice consistency, are affected specifically by value-free random exploration but not by other exploration strategies; Figure 1—figure supplement 3 and Figure 1—figure supplement 4. Moreover, the frequency of picking the novel bandit is affected mainly by a novelty exploration strategy and to a lower extend by UCB exploration (Figure 1—figure supplement 5).

Based on the reviewers’ suggestion, we now add simulations specific to the short and long horizons (to simulate the latter we allowed all other exploration strategies to increase as well; in Appendix 2—table 7). Our simulations show effects were observed both in the short and long horizon condition. We believe that these new simulations provide additional intuitions regarding the two new exploration heuristics and we mention these simulations in the revised manuscript. Please also see the added simulations and illustrations in Figure 1 —figure supplement 3, Figure 1—figure supplement 4, Figure 1—figure supplement 5, which further expand on our findings.

Results: “Additionally, we simulated the effects of other exploration strategies in short and long horizon conditions (Figure 1—figure supplement 3, Figure 1—figure supplement 4, Figure 1—figure supplement 5). To simulate a long (versus short) horizon condition we increased the overall exploration by increasing other exploration strategies. Details about parameter values can be found in Appendix 2—table 7.”

[Editors' note: further revisions were suggested prior to acceptance, as described below.]

Reviewer #1:Thank you for a largely responsive revision. The paper is much improved. A few points remain:Related to previous point 1, the argument in subsection “Probing the contributions of heuristic exploration strategies” does not seem to be entirely correct. The authors claim that "A second prediction is that choice consistency, across repeated trials, is substantially affected by value-free random exploration." However, consistency can also be affected by the softmax parameter. If β is higher then choice consistency is also lower. Also, I am a little bit confused about the simulation results in Figure 1- figure supplement 2E,F. Do both models predict that the consistency of selecting the low value bandit is higher than the consistency of selecting the high value bandit? In line with the argument that higher β also lead to more stochastic choices, I also wonder if that can be the reason why UCB and UCB+ϵ are not that much different in likelihood.

We apologise for a lack of clarity on this point. A key feature of value-free random exploration is that it ignores all information, leading to a completely random selection among the choice options. None of the other exploration strategies shows a similarly strong effect on consistency, because they are still guided by the value of the options, and are disposed to choose the most valuable alternative (in the model’s eye).

The sentence in subsection “Probing the contributions of heuristic exploration strategies”, referred to by the referee, was meant to highlight a comparison to complex exploration strategies, such as directed exploration which consistently explores the less known option. We have now revised this passage to explain what we mean better.

Regarding Figure 1—figure supplement 2E,F, we believe that the referee has misunderstood this figure. We would highlight that the simulations we present here refer to the consistency across all bandits (as consistency is the inverse of switching between bandits). The x-axis shows the simulation of two different levels of our parameters (𝛽, 𝜖, respectively). We agree that this should have been more clear and we have now revised the figure legend to avoid any misunderstanding by the readership. As the reviewer can see, the 𝛽 parameter also has an effect on the consistency, but to a lesser degree, because this clearly prefers medium-valued bandits over low-valued bandits (cf. Figure 1—figure supplement 2C,D).

We agree with the reviewer that this parameter may capture some of the randomness in the UCB model, which leads to a more similar performance between the two models considered. We think this is an interesting observation and we address this point in more detail in the revised discussion.

Results (subsection “Probing the contributions of heuristic exploration strategies”): “A second prediction is that choice consistency, across repeated trials, is directly affected by value-free random exploration, in particular by comparison to more deterministic exploration strategies (e.g. directed exploration) that are value-guided and thus will consistently select the most informative and valuable options.”

Discussion: “A second exploration heuristic that also requires minimal computational resources, value-free random exploration, also plays a role in our task. Even though less optimal, its simplicity and neural plausibility renders it a viable strategy. Indeed, we observe an increase in performance in each model after adding 𝜖, supporting the notion that this strategy is a relevant additional human exploration heuristic. Interestingly, the benefit of 𝜖 is somewhat smaller in a simple UCB model (without novelty bonus), which probably arises because value-based random exploration partially captures some of the increased noisiness.”

Figure 1—figure supplement 2 legend: “Comparison of value-based (softmax) and value-free (𝜖greedy) random exploration. (a) Changing the softmax inverse temperature affects the slope of the sigmoid while changing the 𝜖-greedy parameter (b) affects the compression of the sigmoid. Conceptually, in a softmax exploration mode, as each bandit's expected value is taken into account, (c) the second best bandit (medium-value bandit) is favoured over one with a lower value (low-value bandit) when injecting noise. In contrast, in an 𝜖-greedy exploration mode, (d) bandits are explored equally often irrespective of their expected value. Both simulations were performed on trials without novel bandit. When simulating on all trials we observe this also has a consequence for choice consistency. (e) Choices are more consistent in a low (versus high) softmax exploration mode (i.e. high and low values of 𝛽 respectively), and similarly (f) choices are more consistent in a low (versus high) 𝜖-greedy exploration mode (i.e. low and high values of 𝜖 respectively). When comparing the overall consistency of the two random exploration strategies, consistency is higher in the value-based mode, reflecting a higher probability of (consistently) exploring the second best option, compared to an equal probability of exploring any non-optimal option (inconsistently) in the value-free mode.”

Regarding previous point 2: Were response time differences between value-free exploration and exploitation trials larger in the long horizon than the short horizon condition (i.e., while there was no main effect of bandit, was there an interaction with horizon or trial within horizon and was there a three-way interaction with drug)? Moreover, the response to the mistake issue is not entirely satisfactory. If participants paid (gradually) less attention in the long horizon, then it would also be expected that they make more mistakes in the long horizon condition only.

We would like to highlight here that our previous analysis only investigated the first choice made in each horizon (in line with all other analyses). As previously presented in Figure 2—figure supplement 1B, there is a substantial decrease in response latencies for the long horizon after the first choice, and this could have confounded the analysis. We have now clarified this. We believe that by focusing on the first choice, our analysis should not be affected by subjects paying ‘gradually less attention’ in the long horizon.

Nevertheless, we have now conducted a three-factor analysis that the reviewer suggests, where we investigated the response latencies of the first choice using factors horizon, drug, and bandit. As in our previous findings, we observed no main effect of either bandit (F(1.71,92.46)=1.203, p=.3, 𝜂^2^=.022), horizon (F(1,54)=.71, p=.403, 𝜂^2^=.013), or drug (F(2,54)=2.299, p=.11, 𝜂^2^=.078).

Additionally, there was no interaction between any of the factors (drug-by-bandit interaction: F(3.42,92.46)=.431, p=.757, 𝜂^2^=.016; drug-by-horizon interaction: F(2,54)=.204, p=.816, 𝜂^2^=.008; bandit-by-horizon interaction: F(1.39,75.01)=.298, p=.662, 𝜂^2^=.005; drug-by-bandit-by-horizon interaction: F(2.78,75.01)=1.015, p=.387, 𝜂^2^=.036). We believe this further strengthens our findings and interpretation, and we have now added this analysis to the revised manuscript.

As the reviewer raised an interesting point about the subsequent choices in the long horizon, we have now conducted a new analysis across all long horizon choices using the three factors of bandit, drug and sample (/choice). As shown in our previous Figure 2—figure supplement 1B, we observed a strong effect of sample ((1.54,86.15)=427.047, p<.001, 𝜂^2^=.884), meaning that the response latencies decrease over time. Interestingly, we also observed an effect of bandit (F(1.61,90.12)=7.137, p=.003, 𝜂^2^=.113), as well as a bandit-by-sample interaction (F(3.33,186.41)=4.789, p=.002, 𝜂^2^=.079). No drug effect or other interaction was observed (drug main effect: F(2,56)=.542, p=.585, 𝜂^2^=.019; drug-by-bandit interaction: F(3.22,90.12)=.525, p=.679, 𝜂^2^=.018; drug-by-sample interaction: F(3.08,86.15)=1.039, p=.381, 𝜂^2^=.036, 𝜂^2^=.078; drug-bybandit-by-sample interaction: F(6.66,186.41)=.645, p=.71, 𝜂^2^=.023). Analysing the bandit-sample effect further (Author response image 1), we found this was driven by faster response times for the high-value (exploitation) bandit at the second choice (high-value bandit vs low-value bandit : t(59)=-5.736, p<.001, d=.917; high-value bandit vs novel bandit: t(59)=-6.24, p<.001, d=.599; bandit main effect: F(1.27,70.88)=27.783, p<.001, 𝜂^2^=.332) and a slower response time for the low-value bandit at the second (low-value bandit vs novel bandit: t(59)=3.756, p<.001, d=.432) and third choice (high-value bandit vs low-value bandit : t(59)=-5.194, p<.001, d=.571; low-value bandit vs novel bandit: t(59)=4.448, p<.001, d=.49; high-value bandit vs novel bandit: t(59)=-1.834, p=.072, d=.09; bandit main effect: F(1.23,68.93)=21.318, p<.001, 𝜂^2^=.276). We believe this reflects that when subjects decide to exploit for the remainder of the samples, they respond more quickly than in a situation where they continue to explore. Given that the response times are slower, rather than faster, than the other bandits does not support a notion of hasty mistakes. We have now added this observation to Appendix 1.

Appendix: “When looking at the first choice in both conditions, no differences were evident in RT in a repeated-measures ANOVA with the between-subject factor drug group and the within-subject factors horizon and bandit (bandit main effect: F(1.71,92.46)=1.203, p=.3, 𝜂^2^=.022; horizon main effect: F(1,54)=.71, p=.403, 𝜂^2^=.013; drug main effect: F(2,54)=2.299, p=.11, 𝜂^2^=.078; drug-by-bandit interaction: F(3.42,92.46)=.431, p=.757, 𝜂^2^=.016; drug-by-horizon interaction: F(2,54)=.204, p=.816, 𝜂^2^=.008; bandit-by-horizon interaction: F(1.39,75.01)=.298, p=.662, 𝜂^2^=.005; drug-by-bandit-by-horizon interaction: F(2.78,75.01)=1.015, p=.387, 𝜂^2^=.036).In the long horizon, when looking at all 6 samples, no differences were evident in RT between drug group in the repeated-measures ANOVA with a between-subject factor drug group, and within subject factors bandits and samples (drug main effect: F(2,56)=.542, p=.585, 𝜂^2^=.019). There was an effect of bandit (bandit main effect: F(1.61,90.12)=7.137, p=.003, 𝜂^2^=.113), of sample (sample main effect: F(1.54,86.15)=427.047, p<.001, 𝜂^2^=.884) and an interaction between the two (bandit-by-sample interaction: F(3.33,186.41)=4.789, p=.002, 𝜂^2^=.079; drug-by-bandit interaction: F(3.22,90.12)=.525, p=.679, 𝜂^2^=.018; drug-by-sample interaction: F(3.08,86.15)=1.039, p=.381, 𝜂^2^=.036; drug-by-bandit-by-sample interaction: F(6.66,186.41)=.645, p=.71, 𝜂^2^=.023). Further analysis (not corrected for multiple comparisons) revealed that the interaction between bandit and sample reflected the fact that when looking at samples individually, there was a bandit main effect in the second sample (bandit main effect: F(1.27,70.88)=27.783, p<.001, 𝜂^2^=.332; drug main effect: F(2,56)=.201, p=.819, 𝜂^2^=.007; drug-by-bandit interaction: F(2.53,70.88)=.906, p=.429, 𝜂^2^=.031) and in the third sample (bandit main effect: F(1.23,68.93)=21.318, p<.001, 𝜂^2^=.276; drug main effect: F(2,56)=.102, p=.903, 𝜂^2^=.004; drug-by-bandit interaction: F(2.46,68.93)=.208, p=.855, 𝜂^2^=.007), but not in the other samples first sample: drug main effect: F(2,56)=1.108, p=.337, 𝜂^2^=.038; bandit main effect: F(2,112)=.339, p=.713, 𝜂^2^=.006; drug-by-bandit interaction: F(4,112)=.414, p=.798, 𝜂^2^=.015; 4^th^ sample: (drug main effect: F(2,56)=.43, p=.652, 𝜂^2^=.015; bandit main effect: F(1.36,76.22)=1.348, p=.259, 𝜂^2^=.024; drug-by-bandit interaction: F(2.72,76.22)=.396, p=.737, 𝜂^2^=.014; 5^th^ sample: drug main effect: F(2,56)=.216, p=.806, 𝜂^2^=.008; bandit main effect: F(1.25,69.79)=.218, p=.696, 𝜂^2^=.004; drug-by-bandit interaction: F(2.49,69.79)=.807, p=.474, 𝜂^2^=.028; 6^th^ sample: drug main effect: F(2,56)=1.026, p=.365, 𝜂^2^=.035; bandit main effect: F(1.05,58.81)=.614, p=.444, 𝜂^2^=.011; drug-by-bandit interaction: F(2.1,58.81)=1.216, p=.305, 𝜂^2^=.042). In the second sample, the high-value bandit was chosen faster (high-value bandit vs low-value bandit: t(59)=-5.736, p<.001, d=.917; high-value bandit vs novel bandit: t(59)=-6.24, p<.001, d=.599) and the low-value bandit was chosen slower (low-value bandit vs novel bandit: t(59)=3.756, p<.001, d=.432). In the third sample, the low-value bandit was chosen slower (high-value bandit vs low-value bandit: t(59)=-5.194, p<.001, d=.571; low-value bandit vs novel bandit: t(59)=4.448, p<.001, d=.49; high-value bandit vs novel bandit: t(59)=-1.834, p=.072, d=.09).”

Regarding previous point 8, it is great that the authors followed our suggestion to simulate all models in both the short and long horizon. However, these figures (Figure 1—figure supplement 3 to Figure 1—figure supplement 5) seem to be somewhat confusing. The problem may lie in the parameters selected for simulation. According to Appendix 2 Table 7, the multiple parameters were varied among different models. But I thought they should keep most consistent and only vary the interesting one. For example, shouldn't 𝜂 be kept same or even be zero in the value free random exploration model to show how choices are vary as function of 𝜖? I think the numbers are selected such that the predictions favor the value free random exploration model. If, as authors said, UCB + 𝜖 + 𝜂 is almost as good as Thompson-sampling + 𝜖 + 𝜂, I don’t see what the predictions can be such dramatically different. That is how I interpret the statement that " For simulating the long (versus short) horizon condition, we assumed that not only the key value but also the other exploration strategies increased, as found in our experimental data." Anyway, I feel the simulation data is somehow misleading and need more explanation.

We thank the reviewer for appreciating our effort and apologise that we were not sufficiently clear in explaining what these simulations illustrate. The key purpose of these figures is to illustrate how different aspects of the models affect bandit choices.

In the original reviews, the reviewer suggested to simulate both short and long horizon, and the effects of low and high exploration of a specific exploration strategy. For the low and high exploration, we indeed kept all parameters identical bar the parameter of interest, which is what we changed (as suggested by this reviewer). This is what we show in the short horizon.

For illustrating the long horizon, we tried to accommodate the fact that multiple exploration strategies are elevated in our subjects (compared to the short horizon). We thus decided to also increase other exploration strategies, as listed in Appendix 2—table 7, and correctly observed by this reviewer. However, we do agree that this can be confusing and are happy to remove what we label “long horizon” for clarity.

The parameter values for these simulations are well within the range of fitted model parameters. However, because these are primarily to illustrate the effects of the model (rather than align with exact subjects’ behaviours), we have taken somewhat accentuated values that highlight the specific effects of the parameter more clearly. It is important to note that the key purpose of these illustrations is to compare the effect of low and high exploration within each model, rather than comparing the absolute height of the bars. This was not clear enough in the original captions, and we have now entirely revised these captions.

Besides these simulations, we also provide the model simulations of the complete winning model in the original manuscript. This was shown in Figure 5—figure supplement 1. The reviewer indeed raised a relevant point about this, which is that we did not show the same model fit using the second winning model (UCB). We have now performed this simulation using each participants’ fitted model parameters for the second winning model and have added it as a figure to the manuscript (cf. Figure 5 – figure supplement 3). As one can see, both complex models simulations make fairly similar predictions for our data, as we had previously mentioned, but not shown. We hope this now illustrates our previous notion and we have now detailed this in the revised manuscript.

Discussion: “Importantly, these heuristics were observed in all best models (first, second and third position) even though each incorporated different exploration strategies. This suggests that the complex models make similar predictions in our task. This is also observed in our simulations, and demonstrates that value-free random exploration is at play even when accounting for other value-based forms of random exploration (1, 7), whether fixed or uncertainty-driven.”

Figure 1—figure supplement 3 legend: “Simulation illustrations of high and low exploration on the frequency of picking the low-value bandit using different exploration strategies shows that (a) a high (versus low) value-free random exploration increases the selection of the low-value bandit, whereas neither (b) a high (versus low) novelty exploration, (c) a high (versus low) Thompson-sampling exploration nor (d) a high (versus low) UCB exploration affected this frequency. To illustrate the long (versus short) horizon condition, we accommodated the fact that not only key values but also other exploration strategies were enhanced by increasing multiple exploration strategies, as found in our experimental data (cf. Appendix 2 —table 7 for parameter values). Please note that the difference between low and high exploration is critical here, rather than a comparison of the absolute height of the bars between strategies (which is influences in the models by multiple different exploration strategies). For simulations fitting participants’ data, please see Figure 5—figure supplement 1 and Figure 5—figure supplement 3.”

Figure 1—figure supplement 4 legend: “Simulation illustrations of high and low exploration choice consistency using different exploration strategies shows that (a) a high (versus low) value-free random exploration decreases the proportion of same choices, whereas neither (b) a high (versus low) novelty exploration, (c) a high (versus low) Thompson-sampling exploration nor (d) a high (versus low) UCB exploration affected this measure. To illustrate the long (versus short) horizon condition, accommodated the fact that not only the key value but also other exploration strategies were enhanced by increasing multiple exploration strategies, as found in our experimental data (cf. Appendix 2—table 7 for parameter values). Please note that the difference between low and high exploration is critical here, rather than a comparison of the absolute height of the bars between strategies (which is influences in the models by multiple different exploration strategies). For simulations fitting participants’ data, please see Figure 5—figure supplement 1 and Figure 5—figure supplement 3.”

Figure 1—figure supplement 5 legend: “Simulation illustrations of high and low exploration on the frequency of picking the novel bandit using different exploration strategies shows that (a) a high (versus low) value-free random exploration has little effect on the selection of the novel bandit, whereas (b) a high (versus low) novelty exploration increases this frequency. (c) A high (versus low) Thompson-sampling exploration had little effect and (d) a high (versus low) UCB exploration affected this frequency but to a lower extend than novelty exploration. To illustrate the long (versus short) horizon condition, we accommodated the fact that not only the key value but also other exploration strategies were enhanced by increasing multiple exploration strategies, as found in our experimental data (cf. Appendix 2—table 7 for parameter values). Please note that the difference between low and high exploration is critical here, rather than a comparison of the absolute height of the bars between strategies (which is influences in the models by multiple different exploration strategies). For simulations fitting participants’ data, please see Figure 5—figure supplement 1 and Figure 5—figure supplement 3.”

Reviewer #2:The authors addressed all my comments and made substantial revisions that have strengthened the overall manuscript. Specifically, the new information in Appendix Table 4 with each model's performance and additional analyses on of the other "(close to best)" models further strengthens the authors' claim. The authors also clarified the results on heart rate, RT and PANAS questionnaire, providing additional results and discussing appropriately potential caveats. Further additions in the Discussion address potential mechanisms of propranolol on decision making.

We thank this reviewer for a very positive endorsement of our revisions and for acknowledging our additional analyses have strengthened the paper. We have now addressed the remaining point below.

The only comment I have relates to the sentence that follows (in the Discussion): “In particular, the results indicate that under propranolol behaviour is more deterministic and less influenced by “task-irrelevant” distractions. This aligns with theoretical ideas, as well as recent optogenetic evidence (32), that propose noradrenaline infuses noise in a temporally targeted way (31). It also accords with studies implicating noradrenaline in attention shifts (for a review cf. (76)). Other theories of noradrenaline/catecholamine function can link to determinism (64, 65), although the hypothesized direction of effect is different (i.e. noradrenaline increases determinism)." Here, it is unclear to me how the authors define determinism and how either increasing or decreasing noradrenaline can increase determinism?

Firstly, we apologise for the unclear sentence and confusing wording. We chose “determinism” as the opposite of “stochasticity” (as also capture in our value-free random exploration), but we agree that the term is confusing and have therefore decided to use the latter, more clear terminology.

The directionality of this effect is indeed interesting, and to our understanding not entirely clear.

Theoretical accounts are somewhat contradictory, with a gain-modulation account (Aston-Jones and Cohen, 2005; Servan-Schreiber et al., 1990) suggesting a decrease of stochasticity with increasing noradrenaline function. On the other hand, other theoretical accounts (Dayan and Yu, 2006) suggest that noradrenaline can induce stochasticity. Our findings, showing a reduction in stochasticity after propranolol, favour the latter. However, several other aspects of noradrenaline functioning may explain differential theoretical accounts. They are likely to capture different aspects of the assumed U-shaped noradrenaline functioning curve, and/or they may be relevant in distinct activity modes, such as tonic and phasic firing (cf. Aston-Jones and Cohen, 2005). We have now implemented this discussion in more detail in the revised manuscript.

Discussion: “In particular, the results indicate that under propranolol behaviour is less stochastic and less influenced by “task-irrelevant” distractions. […] Further studies can shed light on how different modes of activity affect value-free random exploration.”